# Multi-level evidence of an allelic hierarchy of *USH2A* variants in hearing, auditory processing and speech/language outcomes

Peter A. Perrino [1,2], Lidiya Talbot[3], Rose Kirkland[4,5], Amanda Hill[6], Amanda R. Rendall[1,2], Hayley S. Mountford[3], Jenny Taylor[4,7], WGS500 Consortium, Alexzandrea N. Buscarello[1,2], Nayana Lahiri[8], Anand Saggar[8], R. Holly Fitch [1,2✉] & Dianne F. Newbury [3✉]

Language development builds upon a complex network of interacting subservient systems. It therefore follows that variations in, and subclinical disruptions of, these systems may have secondary effects on emergent language. In this paper, we consider the relationship between genetic variants, hearing, auditory processing and language development. We employ whole genome sequencing in a discovery family to target association and gene x environment interaction analyses in two large population cohorts; the Avon Longitudinal Study of Parents and Children (ALSPAC) and UK10K. These investigations indicate that *USH2A* variants are associated with altered low-frequency sound perception which, in turn, increases the risk of developmental language disorder. We further show that *Ush2a* heterozygote mice have low-level hearing impairments, persistent higher-order acoustic processing deficits and altered vocalizations. These findings provide new insights into the complexity of genetic mechanisms serving language development and disorders and the relationships between developmental auditory and neural systems.

[1] Department of Psychological Science/Behavioral Neuroscience, University of Connecticut, Storrs, CT, USA. [2] UConn Institute of Brain and Cognitive Sciences; UConn Murine Behavioral Neurogenetics Facility, Storrs, CT, USA. [3] Faculty of Health and Life Sciences, Oxford Brookes University, Oxford OX3 0BP, UK. [4] Wellcome Trust Centre for Human Genetics, Roosevelt Drive, Headington, Oxford OX3 7BN, UK. [5] School of Veterinary Medicine and Science, University of Nottingham, Sutton Bonington Campus, Leicestershire LE12 5RD, UK. [6] Population Health Sciences, Bristol Medical School, University of Bristol, Bristol BS8 2BN, UK. [7] NIHR Biomedical Research Centre, John Radcliffe Hospital, Headley Way, Headington, Oxford OX3 9DU, UK. [8] Institute of Molecular and Clinical Sciences, St George's, University of London & St George's University Hospitals NHS Foundation Trust, London, UK. A full list of members and their affiliations appears in the Supplementary Information. ✉email: roslyn.h.fitch@uconn.edu; diannenewbury@brookes.ac.uk

Learning to use and understand language requires the coordinated development of a whole host of underlying skills and processes. Assuming an initial bottom-up framework as posited by Tallal & others[1], the first step in typical language development requires hearing, which allows infants to identify and parse meaningful signals from background noise and environmental distractors (auditory perception). These sounds are then mapped onto speech units (phonemes) enabling the extraction of meaningful linguistic constructs (comprehension). Even newborn babies turn their heads towards familiar voices and show a preference for speech over white noise[2]. These inclinations are fined-tuned over early life such that, at 3 months, infants show a preference for their native tongue over foreign languages and other vocal sounds (e.g. laughter)[3]. These behavioural distinctions are mirrored at the cognitive level; many studies show that speech evokes distinct neural responses, indicating that language is subject to privileged processing[4]. Early speech processing skills correlate with later language milestones[5] suggesting that they modulate successful language acquisition.

While hearing is not strictly essential to successful language development (see ref. [6]), overt disturbance of the contributory processes can directly disrupt language development; congenital hearing loss is associated with widespread cognitive deficits in domains including attention, memory and language[7]. Less obvious disturbances of audition can also indirectly impact spoken language development, especially if they occur during critical time periods or in addition to other insults. For example, auditory processing disorder (APD), characterized by poor central sound processing despite apparently normal hearing, can lead to difficulties understanding speech in noisy environments[8] and, when persistent, is associated with an increased risk of Developmental Language Disorder (DLD) and Attention Deficit, Hyperactivity Disorder (ADHD)[9]. In the current study we test this framework by assessing whether typical variations across genes that function in auditory pathways may form part of a complex risk mechanism in the emergence of language disorders. In support of our proposed framework, subtle disturbances of auditory processing have been described across many neurodevelopmental disorders where hearing is unaffected (e.g. autistic disorder (as reviewed by ref. [10], ADHD[11] and dyslexia[12]). Similarly, declines in later-life auditory processing skills are correlated with altered language function[13].

The observed overlaps and comorbidities between neurodevelopmental and language disorders illustrate the complexities of contributory networks and the inter-reliance of developmental psychopathologies. In the long-term, all of these disorders are associated with academic limitation and reduced quality of life[8,14] and represent an economic burden upon health-care and education systems[15].

Although APD and DLD are both common childhood conditions (as high as 10%)[16,17], our understanding of the relationships between hearing, auditory processing and language are confounded by these comorbidities alongside a lack of consistent diagnostic guidelines and failure to identify causal mechanisms[18]. As for many neurodevelopmental disorders, genetic contributions are poorly understood[6,19]. While monogenic forms of language disorder have been identified[20], the majority of genetic risk factors are expected to function within complex models where each variant has only a small effect size and is influenced by additional genetic and environmental interactors[19]. Genome-wide screens indicate that large sample sets will be required to map variants that contribute to language disorder[21] and acquisition[22] and recent studies indicate shared genetic effects across behavioural subsets and between disorder and typical development[19].

Current research suggests that genetic effects can overlap between syndromes that have traditionally been considered as clinically distinct. Similarly, genetic overlaps are reported between Mendelian forms of disease and more complex forms of disorder[23–25]. These changes in the field led us to apply an alternative approach to the investigation of molecular mechanisms underlying hearing and language within the current study. More specifically, we investigate the molecular overlaps between hearing, auditory processing and language through the targeted study of a gene that has an established role in audition—USH2A. This gene encodes the Usherin protein, which acts as a lateral link between stereocilium, providing structural organization for hair cell bundle development[26]. Homozygous pathogenic changes in USH2A, and therefore complete absence of the usherin protein, result in disorganization or loss of cochlear outer hair cells[27], leading to congenital hearing loss clinically described as Usher Syndrome (OMIM#276901)[28,29]. This syndrome is a monogenic recessive disorder (4 in 100,000 births) associated with hearing loss specific to the high-frequency ranges, often accompanied by retinitis pigmentosa. The disorder splits into three clinical categories (Types I–III), relating to severity and age of onset[28,30] and homozygous USH2A mutations result in the majority of Type-II cases in which diagnosed individuals are born with hearing loss and develop retinitis pigmentosa at the onset of puberty, yet do not experience the vestibular dysfunction (i.e. difficulties with balance and coordination) found in other types of Usher[31]. In accordance with currently accepted genetic models, individuals with a heterozygous USH2A mutations are considered to be unaffected carriers.

In the current study, we employ whole-genome sequencing in a discovery family to target association and gene x environment interaction analyses in two large population cohorts; the Avon Longitudinal Study of Parents and Children (ALSPAC) and UK10K. We complement these statistical methodologies with the characterization of a mouse model that shows behavioural effects of a heterozygous knockout on hearing, complex acoustic processing, and communicative vocalization. We propose the existence of an "alleleic hierarchy" of variation within which different variant types have divergent effects upon auditory thresholds. We show that USH2A variants exert direct effects upon low-frequency hearing and indirectly affect the risk of language disorder through the modulation of subsequent auditory perception and language development. These findings suggest a shared genetic etiology between hearing mechanisms, central auditory processing and language development and support the targeted investigation of sub-serving mechanisms in relation to language development.

## Results

**An USH2A variant cosegregates with language disorder**. A multi-generational family was ascertained for genetic investigation (Fig. 1). The family included seven Non-Founder individuals, all of whom were affected by a severe expressive language disorder. Affected individuals presented with slow and dysfluent speech and difficulties characteristic of APD, namely processing speech and following instructions, particularly in the presence of background noise or absence of visual cues (e.g. on the phone). The proband (IV.I, Fig. 1) and her sister (IV.2, Fig. 1) attended a special school for children with speech and language difficulties. Hearing assessments in the proband and sister were normal. Pure-tone audiometry tests in the Grandfather (II.I, Fig. 1) indicated normal hearing thresholds but deficits were noted across three tests of central auditory processing (dichotic digits, frequency pattern and duration pattern). Further details of the phenotype are provided in the Methods. Genome sequencing of two individuals (II.2 and IV.1) identified a novel stop-gain mutation (a change which results in a truncated protein) in the USH2A gene (NP_996816:p.Gln4541*) shared by all affected

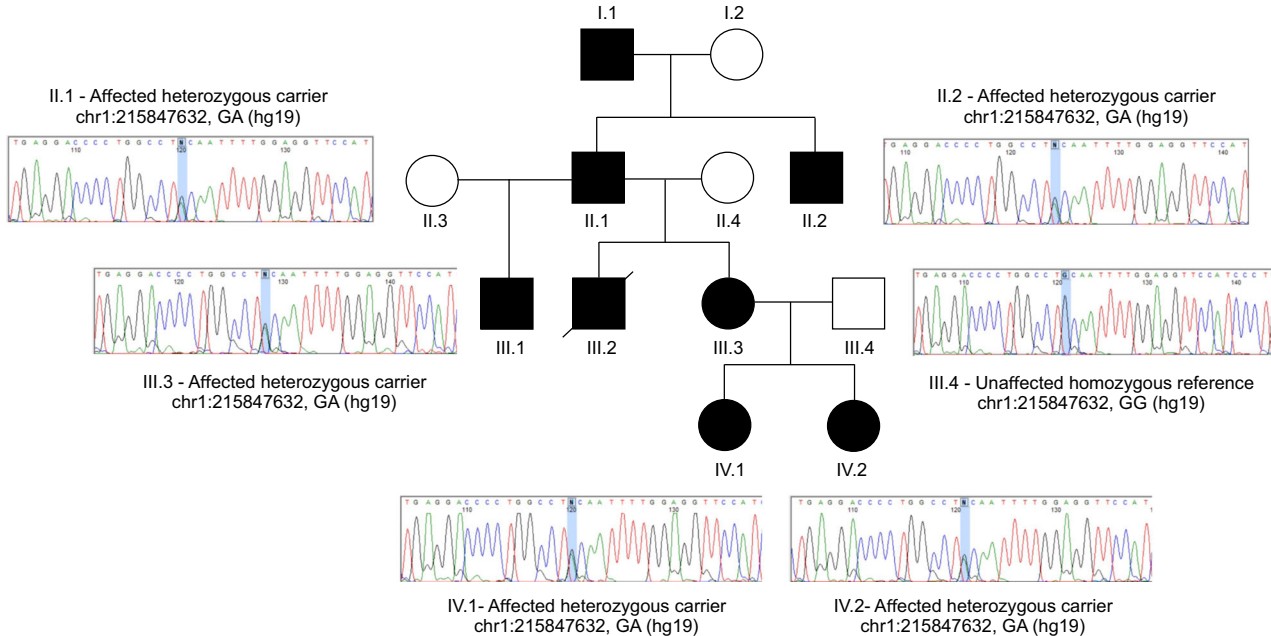

**Fig. 1 Discovery Family and cosegregation of *USH2A* variant.** Individuals affected by APD are coloured in black. Unaffected individuals are shown in white. Line through symbol indicates that individual is deceased. All descendants of individual I.1 were affected by a severe expressive language disorder. The only unaffected individuals in the family were incoming Founder members (I.2, II.2, II.3, III.4). Chromatograms show validation of *USH2A* variant in wider family members. The variant (position indicated by a blue highlight bar) was observed in a heterozygous form in all affected individuals.

family members (Fig. 1, Supplementary Dataset 1). This variant (rs765476745) has previously been documented in cases of Usher syndrome and retinitis pigmentosa in whom it was documented to occur with a secondary pathogenic variant in a compound heterozygote mechanism[32–34]. The hearing of heterozygotes was not assessed within these previous studies[32–34]. rs765476745 has a Combined Annotation Dependent Depletion (CADD) score of 40, placing it in the top 0.01% of deleterious variants in the genome[35], and meets clinical guidelines from the American College of Medical Genetics (ACMG) for a "pathogenic variant"[36]; it confers a stop mutation (Very strong evidence of pathogenicity—PVS1), has previously been described as pathogenic (Strong evidence of pathogenicity—PS1), is absent from population databases (Moderate evidence of pathogenicity—PM2) and cosegregates with disease (Supporting evidence of pathogenicity—PP1)[36]. No secondary putative pathogenic changes were found in other genes related to Usher syndrome or hearing loss (Supplementary Dataset 1). Given the inheritance pattern and complete cosegregation, in the absence of a second pathogenic variant, we hypothesized that the observed heterozygous loss of USH2A could account for the observed language disorder in this family.

**Heterozygous *Ush2a* KOs have altered auditory perception.** To elucidate the behavioural effects of heterozygous *Ush2a* loss, we generated heterozygous (HT) and full knockout (KO) mice. Hearing and complex acoustic discrimination thresholds were compared against WT controls on a series of prepulse inhibition (PPI) tasks[37]. Initially, using a simple single-tone detection task at 40 kHz (high-frequency), individual ANOVAs revealed that KO mice trended to the expected hearing impairment typical of Usher syndrome, while HT mice performed similarly to WT controls [(Overall): $F(2,32) = 1.995$, $p = 0.153$, one-tail; (WT vs. HT): $F(1, 22) = 0.619$, $p = 0.440$; (WT vs. KO): $F(1, 21) = 3.125$, $p = 0.092$; (HT vs. KO): $F(1, 21) = 1.517$, $p = 0.232$ (Fig. 2a). In contrast, using this same task at 15 kHz (low-frequency), HT mice performed significantly worse than KOs and trended to worse than

WTs [(Overall): $F(2,32) = 3.697$, $p = 0.36$; (WT vs. HT): $F(1, 22) = 3.201$, $p = 0.087$; (WT vs. KO): $F(1, 21) = 1.054$, $p = 0.316$; (HT vs. KO): $F(1, 21) = 5.016$, $p = 0.036$], while WT and KO did not differ (Fig. 2a). In order to assess possible higher-order processing deficits, individual scores on the above single-tone task were used as covariates (frequency-matched) to analyze more complex PPI measures (thus eliminating variance due to hearing impairments). In repeated measures ANCOVAs with Genotype as the between-subjects variable and Day and Cue as the within-subject variables, deficits were again evident for HT mice on complex low-frequency tasks [Embedded Tone 100: 10.5 kHz: $F(2, 31) = 3.691$, $p = 0.036$; Embedded Tone 10: 10.5 kHz: $F(2, 31) = 4.635$, $p = 0.017$], and for KO mice on higher frequency tasks [Pitch Discrimination: 40.5 kHz (WT vs. KO); $F(1, 20) = 9.232$, $p = 0.006$] (Fig. 2b–d). These findings indicate that *Ush2a*-mediated perceptual deficits include higher-order dysfunction, even when variance due to hearing loss was removed.

**Heterozygous *Ush2a* KOs have altered vocalizations.** Given the reported comorbidity between auditory processing and language impairments and the presence of dysarthria in the discovery family, we investigated whether *Ush2a* knockout in mice altered the properties of their ultrasonic vocalizations. Results showed that *Ush2a* HT mice vocalized at significantly higher frequencies (pitch) across most syllable types [$F(2, 16100) = 83.476$, $p = 0.000$] and produced calls that were shorter and louder than WTs [duration: $F(2, 16100) = 26.70$, $p = 0.000$; volume: $F(2, 16100) = 142.54$, $p = 0.000$] (Fig. 3). (See Supplementary Fig. 1 for images and coding technology for eight primary call types assessed). Interestingly, *Ush2a* KO mice also produced higher pitched calls suggesting that disruption to auditory processing ability (regardless of the frequency of the stimuli) results in impaired expressive communication ability. This could reflect the importance of intact auditory feedback for vocal development (e.g. anomalous song production in deafened birds), and is consistent with the higher vocal pitch observed in in profoundly deaf speakers[38]. The putative social impact of any vocalization

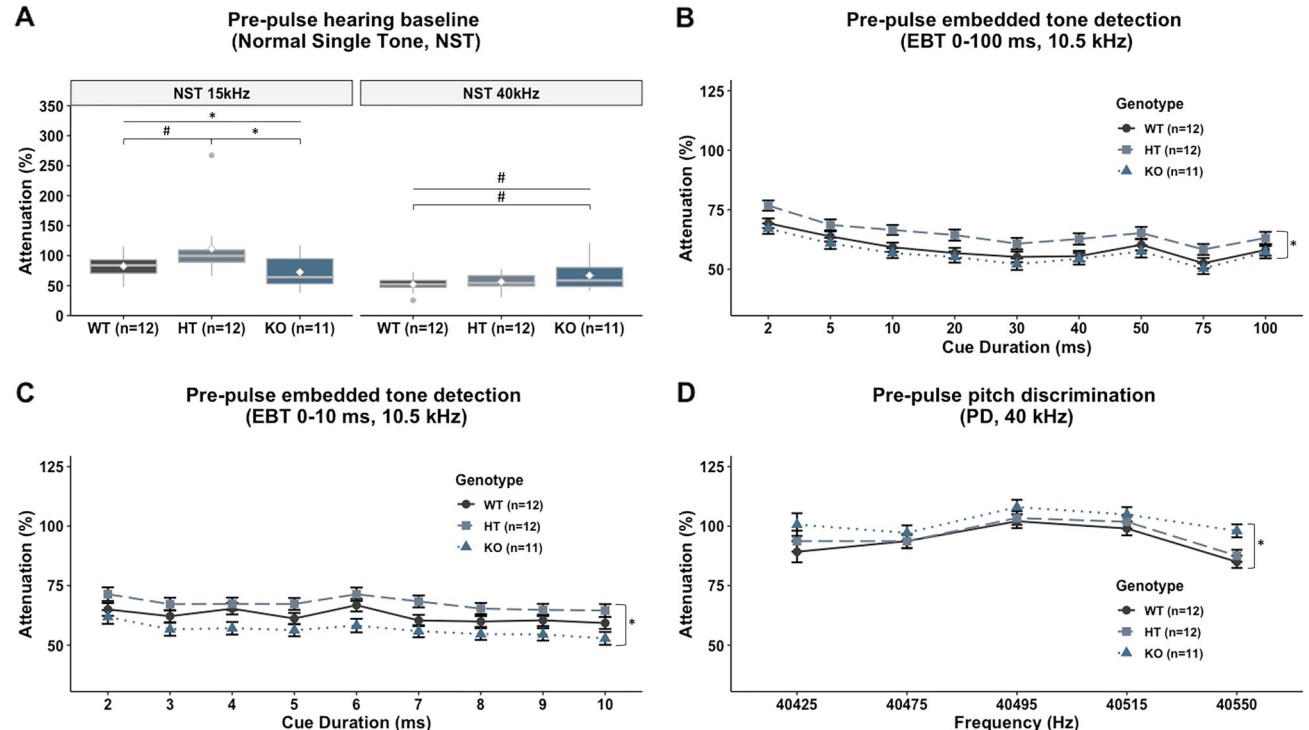

**Fig. 2 *Ush2a* HT mice show low-frequency auditory impairments.** Relative performance on various prepulse inhibition paradigms (lower scores equal better performance). **a** Normal Single Tone at 15 kHz and 40 kHz displayed. **b** Embedded Tone 0-100 at 10.5 kHz analyzed across days using NST 15 kHz as a covariate. **c** Embedded Tone 0-10 at 10.5 kHz analyzed across days using NST 15 kHz as a covariate. **d** Pitch Discrimination at 40 kHz analyzed across days using NST 40 kHz as covariate. Data shown are mean ± SEM for each Genotype. *$p < 0.05$; #$p < 0.15$. White diamond indicates Genotype mean. All panels included 35 biologically independent animals (12 WT, 12 HT, 11 KO). Data underlying these figures are provided in Supplementary Dataset 3.

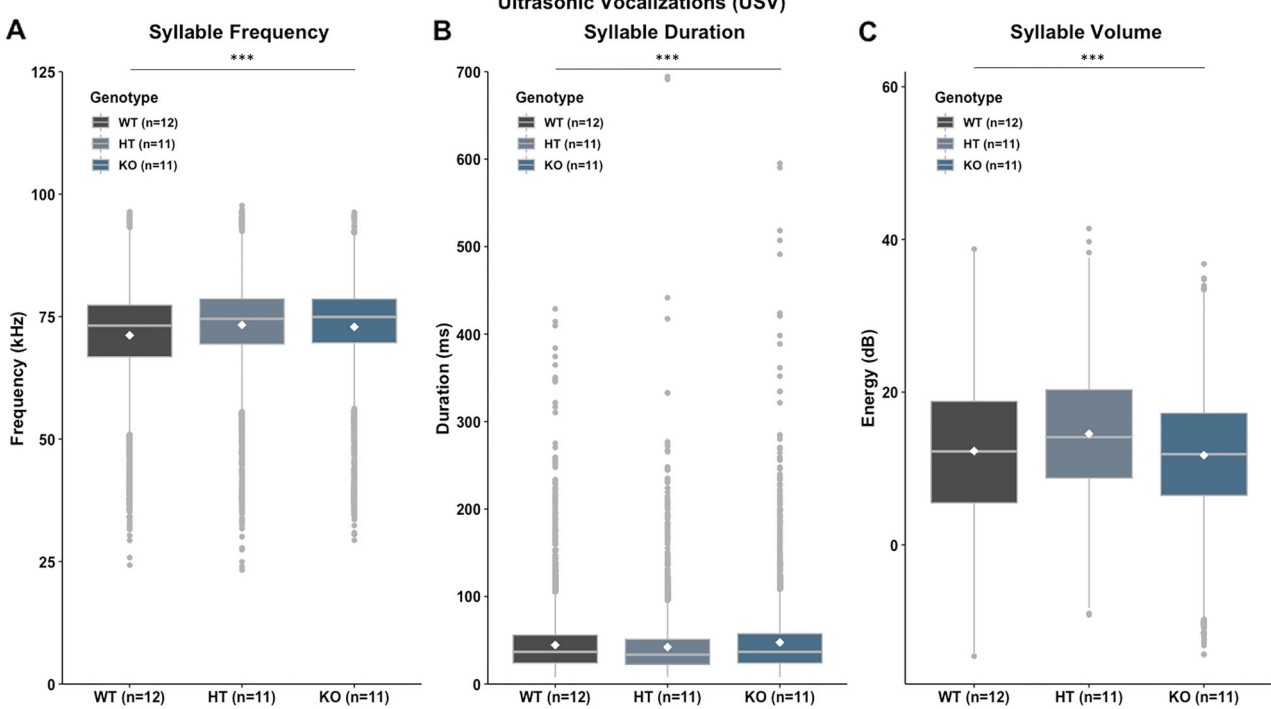

**Fig. 3 *Ush2a* HT mice produce altered ultrasonic vocalizations (USVs). a** Syllable Frequency per Genotype collapsed across syllable category. **b** Syllable Duration per Genotype collapsed across syllable category. **c** Syllable Volume per Genotype collapsed across syllable category. Data shown are mean ± SEM for each Genotype. ***$p < 0.001$. White diamond indicates Genotype mean. All panels included 34 biologically independent animals (12 WT, 11 HT, 11 KO). Data underlying these figures are provided in Supplementary Dataset 4.

**Table 1 Clinically relevant mutations observed in the UK10K dataset.**

| Genome Location (hg19) | SNP ID | Ref | Alt | No. of carriers identified | QUAL score | MAF (gnomAD) | Functional effect | DNA change | Amino acid change |
|---|---|---|---|---|---|---|---|---|---|
| chr1:215956104 | rs111033264 | A | G | 1 | 24.8 | 2.889E-07 | Missense | c.10561T > C | p.Trp3521Arg |
| chr1:215963510 | rs148660051 | C | T | 4 | 247 | 0.000003829 | Missense | c.10073G > A | p.Cys3358Tyr |
| chr1:216019240 | rs397518041 | C | T | 1 | 46.4 | 0.00001627 | Stop-Gain | c.8981G> A | p.Trp2994* |
| chr1:216420436 | rs80338903 | C | – | 7 | 967 | 0.0006835 | Frameshift | c.2299delG | p.Glu767Serfs |
| chr1:216497582 | rs121912600 | C | A | 1 | 171 | 0.00004079 | Missense | c.1256G > T | p.Cys419Phe |

Genome location is given in the format chr:variant position (hg19). QUAL score refers to PHRED quality score for sequence data at given base. Ref is the allele observed in the Human Reference sequence (hg19). Alt is the alternative allele observed in the UK10K. All variants were observed in a heterozygous form. Minor Allele Frequency (MAF) is given for all samples in the gnomAD population database (https://gnomad.broadinstitute.org/). DNA change refers to the position in the coding sequence for the gene. Amino acid change refers to the position altered in the protein.

anomalies will require additional study. Our behavioural investigations of the mouse model indicate that heterozygous disruption of *Ush2a* leads to altered low-frequency hearing thresholds, a phenotype distinct from the high-frequency hearing loss of Usher syndrome (and replicated in null *Ush2a* mice). Moreover, these altered low-frequency thresholds were further associated with higher-order auditory processing deficits, as well as disrupted vocalizations.

**Pathogenic *USH2A* carriers recapitulate the mouse model**. The description of a single case family does not warrant a claim of causation, even when supported by an animal model. We therefore sought to characterize the developmental profiles of other individuals with heterozygous *USH2A* knockout. The effects of pathogenic *USH2A* variants were explored through the investigation of developmental profiles of UK10K children[39]. Fourteen UK10K individuals (12 M:2 F, from 1646 individuals with sequence and phenotypic data available, 0.85%, Table 1) were identified as carriers of *USH2A* changes that were designated as "pathogenic" in ClinVar. These consisted of five distinct variants which were always detected in a heterozygous form (Table 1). The variant found in the discovery family (rs765476745) was not present in the UK10K samples (Table 1). Analyses of developmental behavioural data showed that *USH2A* carriers scored below expected on measures of early vocabulary (Cohen's d = 0.7237, 95% CI = 0.18–1.27) and word combinations (Cohen's d = 1.09, 95% CI = 0.54–1.63) (Table 2). Parents of carriers were twice as likely to be concerned about their child's speech at 3 years of age (RR = 2.07, 95% CI = 0.57–7.49) and reported a higher incidence of stuttering (RR = 2.92, 95% CI = 1.05–8.08) and dyslexia (RR = 1.88, 95% CI = 0.28–12.45) at age 8 (Table 3). In support of our mouse model, we also observed a low-frequency-specific hearing phenotype in heterozygous individuals; carriers had average low-frequency (500 Hz) hearing thresholds 1.2 dB HL above those of non-carriers (Table 2) but did not display overt hearing loss (Table 3) or differences at higher frequencies (Table 2). Thus across mouse and human data, we find evidence that heterozygous *USH2A* variants affect higher-order auditory processing and increase the risk of delayed language milestones. In line with current literature, these variants *alone* do not result in a discernible carrier phenotype (as would be expected in a monogenic model)[40]. Instead, we propose that they form part of a genetic risk factor within a complex genetic model.

**USH2A variants are associated with low-frequency hearing**. Targeted association analyses were performed to investigate the wider effects of *USH2A* polymorphisms on hearing and language within typical development. SNPs tagging common *USH2A* variants were analysed for allelic association in the ALSPAC cohort[41] (N = 7691 individuals, N = 127 variants). Three measures of

language (early vocabulary size (vocab), nonword repetition (NWR) and developmental language disorder (DLD) status), and two hearing measures (low-frequency hearing (minimum air conduction threshold at 0.5 kHz; MinLow) and mid-frequency hearing (minimum air conduction threshold at 1, 2 and 4 kHz; MinMid)) were assessed. In support of the carrier and mouse findings described above, association was observed for a cluster of SNPs located between exons 4 and 12 of *USH2A*, specifically with low-frequency hearing (Table 4). The top-associated SNP (rs10864237, minP = $6.9 \times 10^{-5}$) explained 0.3% of variance in low-frequency hearing thresholds (βSE = 0.13) representing a 1 dB HL difference between risk (TT genotype) and non-risk (CC genotype) individuals. These analyses therefore extend our previous findings to encompass common variants in *USH2A* and low-frequency hearing thresholds within the typical range.

**Gene-environment interactions associate with language**. We then explored a hypothesis that *USH2A* modulates low-frequency hearing thresholds through indirect modulatory effects on subsequent auditory perception and language development. In this model, common *USH2A* variants impact hearing but also exert secondary impacts on speech and language development, presumably as a result of less effective higher-order auditory perception. Specifically, we again assessed common variants in *USH2A* for association to language outcomes in ALSPAC, but this time included low-frequency hearing thresholds as an interaction factor[42]. Association was now observed with early vocabulary (rs7532570) (Table 4). Within this interactive model, rs7532570 had a p-value of $8.6 \times 10^{-5}$ compared to P = 0.15 in the additive model. When combined with the findings of the direct association analyses, these data suggest that common variants in *USH2A* can modify low-frequency hearing thresholds and that, when thresholds are altered *USH2A* can, in turn, modulate the risk of disrupted language development. This mirrors the relationship observed between low-frequency hearing and vocalization in *Ush2a* HT mice, suggesting a parallel modulatory gene-environment (GxE) interaction in mice and together confirming that auditory perception represents a building block for language development.

**Genetic variants in USH2A exert distinct effects**. To gain a more complete picture of *USH2A* variation, gene-based analyses were performed using UK10K genome sequence data (N = 1646 individuals), enabling the combined consideration of rare and common variants across coding and non-coding regions within a single test. This large sequence dataset allows the detection of variants with expected frequency as low at 0.03%. Three variant selection thresholds were considered (all variants (N = 7619), rare variants (MAF ≤ 1%, N = 5424) common variants (MAF ≥ 5%, N = 1335)) in relation to the same three language and two

**Table 2 Quantitative measures of language, reading and cognition in carriers of *USH2A* compared to non-carriers in UK10K dataset.**

| Measure | Age | Range of scores (carriers) | Mean score (carriers) | No. of non-carriers | Range of scores (non-carriers) | Mean Score (non-carriers) | SD (non-carriers) | 5th percentile | No carriers below 5th percentile |
|---|---|---|---|---|---|---|---|---|---|
| Vocabulary score | 3 years | 64–246 | 214.77 | 1601 | 0–246 | 232.42 | 24.81 | 186 | 2 of 13 |
| Plurals score | 3 years | 0–12 | 10.33 | 1594 | 1–12 | 10.32 | 2.02 | 6 | 0 of 12 |
| Past tense score | 3 years | 0–42 | 35.17 | 1584 | 0–42 | 34.23 | 9.43 | 13 | 1 of 12 |
| Word combination score | 3 years | 0–26 | 18.15 | 1593 | 0–26 | 22.79 | 4.31 | 15 | 4 of 13 |
| Reading score: WORD | 7 years | 14–45 | 27.71 | 1558 | 0–50 | 30.11 | 8.74 | 15 | 1 of 14 |
| Spelling score | 7 years | 42064 | 6.93 | 1547 | 0–15 | 8.36 | 4.27 | 2 | 0 of 14 |
| WOLD comprehension | 8 years | 41456 | 8.36 | 1557 | 2–14 | 7.76 | 1.9 | 5 | 0 of 14 |
| Nonword Repetition | 8 years | 43377 | 7.07 | 1558 | 0–12 | 7.47 | 2.46 | 3 | 0 of 14 |
| WISC—Verbal IQ | 8 years | 99–130 | 116 | 1551 | 54–155 | 112.02 | 16.73 | 86 | 0 of 14 |
| WISC—Performance IQ | 8 years | 82–132 | 102.71 | 1550 | 46–145 | 103.56 | 16.87 | 76 | 0 of 14 |
| WISC—Total IQ | 8 years | 90–132 | 111.29 | 1545 | 46–148 | 109.22 | 16.21 | 82 | 0 of 14 |
| Air conduction Right average 0.5, 1, 2, 4 kHz | 7 years | 2.5 to 12.5 | 8.08 | 1241 | −3.75 to 37.5 | 8.18 | 5.31 | 17.5 | 0 of 13 |
| Air conduction Left average 0.5, 1, 2, 4 kHz | 7 years | 0 to 16.25 | 7.79 | 1241 | −8.75 to 40 | 7.90 | 5.45 | 17.5 | 0 of 13 |
| MinMid—minimum air conduction thresholds across the left and right ears, averaged across 1, 2 and 4KHz | 7 years | −1.67 to 11.67 | 5.13 | 1263 | −8.33 to 40 | 6.48 | 5.53 | 16.67 | 0 of 13 |
| MinLow—minimum air conduction thresholds across the left and right ears at 0.5KHz | 7 years | 5 to 15 | 11.15 | 1240 | −10 to 35 | 9.96 | 5.81 | 20 | 0 of 13 |

Fifteen quantitative measures of language and hearing were compared between individuals carrying *USH2A* variants that have previously been reported as pathogenic and control UK10K individuals without such variants.

**Table 3 Discrete measures of educational support, neurodevelopmental disorders and hearing in carriers of *USH2A* compared to non-carriers in UK10K dataset.**

| Measure | Age measured | No affected *USH2A* carriers | Freq in *USH2A* carriers | No. of affected non-carriers | Freq in non-carriers |
|---|---|---|---|---|---|
| Carer worried about child's speech | 3 years | 2 of 13 | 0.15 | 120 of 1580 | 0.08 |
| Child has learning difficulties requiring special arrangements at school | 7 years | 0 of 14 | 0 | 55 of 1548 | 0.04 |
| Child has speech problems requiring special arrangements at school | 7 years | 0 of 14 | 0 | 13 of 1548 | 0.01 |
| Child has hearing problems requiring special arrangements at school | 7 years | 0 of 14 | 0 | 30 of 1548 | 0.02 |
| Child has eyesight problems requiring special arrangements at school | 7 years | 0 of 14 | 0 | 13 of 1548 | 0.01 |
| Child has physical problems requiring special arrangements at school | 7 years | 0 of 14 | 0 | 12 of 1548 | 0.01 |
| Child has reading difficulties requiring special arrangements at school | 7 years | 0 of 14 | 0 | 70 of 1548 | 0.05 |
| Child has emotional/behavioural problems requiring special arrangements at school | 7 years | 0 of 14 | 0 | 22 of 1548 | 0.01 |
| Child stutters/stumbles when speaks | 8 years | 3 of 14 | 0.21 | 119 of 1560 | 0.08 |
| DAWBA DSM-IV clinical diagnosis - Any ADHD disorder | 7 years | 0 of 14 | 0 | 20 of 1569 | 0.01 |
| Mother told child has dyslexia | 9 years | 1 of 12 | 0.08 | 71 of 1540 | 0.05 |
| Mother told child has dyspraxia | 9 years | 0 of 11 | 0 | 29 of 1513 | 0.02 |
| Mother told child has dyscalculia | 9 years | 0 of 11 | 0 | 8 of 1496 | 0.01 |
| Hearing Impairment (AC thresholds greater than 20db HL at 1,2,4 kHz) | 7 years | 0 of 13 | 0 | 114 of 1492 | 0.08 |
| OME/abnormal middle ear pressure (<−100 mm H2O) | 7 years | 5 of 14 | 0.36 | 422 of 1510 | 0.28 |

Fifteen binary measures of developmental difficulties were compared between individuals carrying *USH2A* variants that have previously been reported as pathogenic and control UK10K individuals without such variants.

hearing measures described above. Results showed dichotomous effects between variant frequency classes; association to hearing measures was driven by common variants while DLD status was marginally associated with rare variants (Table 4).

**GxE effects implicate hearing-modulated language pathways.** To explore similar genetic effects at a genome-wide level, a GxE interaction study (GWIs) was completed. These exploratory analyses enabled the identification of common variants that influence language through low-frequency hearing and,

additionally allowed the evaluation of genes implicated in hearing within the model identified[41]. Taking direction from the GxE analyses above, a linear regression model was employed with early vocabulary as the dependent variable and low-frequency hearing thresholds as an interaction term (Supplementary Fig. 2). Eight SNPs reached genome-wide significance ($P < 5 \times 10^{-8}$), while 450 SNPs across 139 HGNC transcripts were nominally associated ($P \le 10^{-5}$) (Supplementary Dataset 2). Pathway analyses did not indicate an enrichment of genes previously related to hearing or language (Supplementary Table 2) but instead revealed an enrichment of protein-binding factors involved in cell

**Table 4 Association analyses of variants across USH2A in relation to language and hearing outcomes.**

| ALSPAC SNP-BASED analysis | | | | Additive model | | | | | Interaction | | |
|---|---|---|---|---|---|---|---|---|---|---|---|
| CHR | SNP | BP (hg19) | A1 | MinLow | MinMid | DLD | NWR | Vocab | DLD | NWR | Vocab |
| 1 | rs682319 | 216,417,675 | T | 0.0274 | 0.0706 | 0.742 | 0.804 | 0.0930 | 0.850 | 0.0914 | **0.00056** |
| 1 | rs11120747 | 216,438,500 | G | 0.0062 | 0.901 | 0.194 | 0.961 | 0.238 | 0.962 | 0.0454 | 0.436 |
| 1 | rs2168924 | 216,440,105 | A | 0.963 | 0.0147 | 0.463 | 0.0537 | 0.349 | 0.525 | 0.0375 | 0.393 |
| 1 | rs1159143 | 216,454,483 | T | 0.0033 | 0.115 | 0.319 | 0.720 | 0.538 | 0.789 | 0.822 | 0.0749 |
| 1 | rs10864237 | 216,466,861 | C | **6.92E-05** | 0.00999 | 0.317 | 0.988 | 0.244 | 0.912 | 0.184 | 0.531 |
| 1 | rs17651066 | 216,470,121 | C | 0.0795 | 0.0676 | 0.493 | 0.315 | 0.122 | 0.250 | 0.309 | 0.0022 |
| 1 | rs7532570 | 216,504,269 | G | 0.407 | 0.410 | 0.848 | 0.835 | 0.150 | 0.813 | 0.266 | **8.60E-05** |
| 1 | rs1606357 | 216,521,091 | T | 0.00022 | 0.0425 | 0.780 | 0.871 | 0.491 | 0.358 | 0.611 | 0.532 |
| 1 | rs17657634 | 216,552,571 | G | 0.910 | 0.153 | 0.337 | 0.0317 | 0.728 | 0.590 | 0.917 | 0.227 |
| 1 | rs4253963 | 216,592,003 | T | 0.0146 | 0.0689 | 0.734 | 0.617 | 0.160 | 0.301 | 0.688 | 0.383 |
| 1 | rs10779261 | 216,595,306 | C | 0.0340 | 0.0534 | 0.863 | 0.825 | 0.676 | 0.283 | 0.582 | 0.885 |
| 1 | rs12723493 | 216,605,071 | A | 0.0187 | 0.233 | 0.393 | 0.659 | 0.827 | 0.650 | 0.166 | 0.311 |

**UK10K GENE-BASED analysis**

| Variant subset | MinLow | MinMid | DLD | NWR | Vocab |
|---|---|---|---|---|---|
| All variants | 0.061 | 0.089 | 0.092 | 0.162 | 0.669 |
| Rare variants (MAF ≤1%) | 0.932 | 0.513 | 0.0094 | 0.313 | 0.282 |
| Common variants (MAF ≥5%) | 0.025 | 0.0087 | 0.710 | 0.204 | 0.277 |

At the variant level, SNPs were directly associated with low-frequency hearing thresholds but not language outcomes within an additive model.
In contrast, in an interactive model, which considers interactions between variants and low-frequency hearing thresholds, association was observed with early language outcomes.
At the gene level, association to hearing measures was largely driven by common variants while association to language impairment was stronger with rare variants.
SNPs are shown for 5′ region of association only (chr1:216,438,500-216,521,091, hg19).
Bold values indicate that p-value was significant after a Bonferroni correction for multiple testing.

adhesion and cellular movement (Table 5). Cellular components of cell projections, lamellipodia and synapses were also over-represented (Table 5). These genome-wide analyses therefore implicate cell migration and connectivity as potential mechanisms for the underlying effect of auditory perception upon speech and language development.

## Discussion

Language development is a multifaceted trait that relies on interactions between many sub-servant mechanisms each subject to genetic, cognitive and environmental effects, including auditory processing and hearing. In this study, we consider developmental links between a specific candidate gene (USH2A), hearing, auditory perception, communicative mouse vocalization and human vocabulary. Together, our data provide evidence that auditory perception represents a building block for communicative and language development. The identification of a USH2A stop-gain variant in the discovery family was substantiated by behavioural investigation of heterozygous Ush2a knockout mice (Ush2a$^{+/-}$). In contrast to full knockouts, these mice presented with a distinctive low-frequency hearing loss ($p < 0.05$ at 15 Hz), accompanied by impairments in complex sound processing that was present even after variance due to hearing loss was removed, and also altered ultrasonic vocalizations. Population data corroborated the functional effects of USH2A in audition and early language development; children in the UK10K cohort[39] who carried pathogenic variants had increased low-frequency hearing thresholds (+1.2 dB HL at 500 Hz) and showed reduced early vocabulary when compared to non-carriers. In a cohort of typically developing individuals[41], variants at the 5′ end of the gene were directly associated with increased low-frequency hearing thresholds (minP = $6.9 \times 10^{-5}$). Within an interactive genetic model, individuals carrying risk variants in the presence of altered low-frequency hearing thresholds were found to have a smaller vocabulary than those who carried only one of these risk factors in isolation (minP = $8.6 \times 10^{-5}$). Together, these data demonstrate that allelic variations in USH2A are associated with altered low-level hearing thresholds that, in turn,

impact speech and language development through the modulation of higher-order acoustic processing. As such, even a subtle degradation in hearing and subsequent complex acoustic processing (as seen in heterozygous Ush2a mice) could developmentally derail language processing in humans.

Our findings are consistent with emerging evidence that different variant types can associate with variable outcomes, forming an "allelic hierarchy" of disease-causing and complex risk variants, representing a shift from Mendelian genetic models[43]. We extend this hypothesis by demonstrating a multifaceted allelic hierarchy in which rare and common variants within the same gene can form reciprocal influences upon gene functions under different environmental influences. The finding that heterozygous disruption of USH2A led to altered hearing thresholds in the low-frequency ranges was unexpected, as complete loss of this gene results in Type-II Usher Syndrome characterized by congenital high-frequency hearing loss[40]. While one previous study suggested that carrier individuals may experience slight hearing disturbances[44], heterozygotes are generally considered aphenotypic and do not show obvious deficits in clinical hearing tests[40]. Our findings provide a molecular explanation for this; heterozygous gene disruptions are typified by subtle changes in the processing of low-frequency sounds that may be incidental to routine audiologic assessment. Such changes would not necessarily be detected in a clinical setting where the focus would be on Usher-related *high*-frequency hearing loss. Notably, we found that the low-frequency thresholds of carrier individuals were consistently (marginally) below those of non-carriers, though still within typical range. While it is unlikely that such subtle changes in hearing thresholds (1–2 dB) at these frequencies would directly lead to language disorder, we propose that mild changes in low-level hearing may exert a snow-ball effect that derails higher-order communicative processing. This model is akin to that described for persistent otitis media with effusion which, in itself, does not cause language disorder but may represent a risk factor when persistent[45]. Our findings are of clinical importance given that heterozygous loss of the USH2A gene is relatively common–we found that carriers of heterozygous pathogenic variants constitute 0.85% of the UK10K cohort studied here. This

**Table 5 Pathway analyses of genes implicated in GxE interaction effects of low-frequency hearing upon vocabulary development.**

| Pathway ID | Pathway description | No. Genes | P (FDR) | Matching genes |
|---|---|---|---|---|
| | | | | *GO processes* |
| GO:0007155 | Cell adhesion | 21 | 0.00254 | APP,BTN3A1,CADM2,CDH18,CLDN11,CLSTN2,CNTNAP2,COL13A1,FAT3,FOXP1,GPR56,ITGA9,ITGBL1,NUAK1,PDGFRA,PRKCA,PRKCE,PTPRK,PTPRM,SEMA5A |
| GO:0051270 | Regulation of cellular component movement | 17 | 0.00842 | ANK2,CAMK1D,DACH1,FOXP1,GPR56,MAGI2,NTRK3,PDGFRA,PLXNA4,PRKCA,PRKCE,PTPRK,PTPRM,SCN5A,SEMA5A,SYNE1,SYNE2 |
| | | | | *GO functions* |
| GO:0005515 | Protein binding | 50 | 0.0382 | AGBL1,ANK2,ANLN,APP,BCL2L1,CDYL,CLDN11,CNTNAP2,CPE,DEPDC5,DLG2,DMRT3,ELMO1,ESR1,ESRRG,EXOC4,FCRL2,FOXP1,GPR56,LRRFIP1,MAGI2,MAP7,MECOM,MICALCL,NTRK3,NUAK1,PAX5,PCCA,PDGFRA,PHACTR4,PRAME,PRKCA,PRKCE,PTPRK,PTPRM,RAB11FIP4,RAD23B,RALYL,SCN5A,SEMA5A,SEPT7,SPPL2C,STXBP6,SYNE1,SYNE2,TLR1,TP73,TRIM33,USH2A,WWOX |
| | | | | *GO cellular components* |
| GO:0071944 | Cell periphery | 58 | 4.87E-06 | ADCY9,ANK2,ANLN,APP,BNC2,BTN3A1,CADM2,CDC42BPB,CDH11,CLSTN2,CNTNAP2,COL13A1,CPE,DLG2,DPP10,ELMO1,EPN1,ESR1,FAT3,FCRL3,GPHN,GPR56,HTR1E,KCNJ1,LGR5,LRRFIP1,LYPD6B,MAGI2,MAP7,MCF2L,MDGA2,MGAM,NTRK3,PDGFRA,PLEK2,PLXNA4,PRAME,PRKCA,PRKCE,PTPRK,PTPRM,PTPRN2,RIMBP2,SCN5A,SEMA5A,SEPT7,SLC22A3,SLC7A14,STXBP6,SYNE1,SYNE2,TLR1,USH2A |
| GO:0005886 | Plasma membrane | 57 | 4.87E-06 | ADCY9,ANK2,APP,BNC2,BTN3A1,CADM2,CDC42BPB,CDH11,CDH18,CDH26,CLDN11,CLSTN2,CNTNAP2,COL13A1,CPE,DLG2,DPP10,ELMO1,EPN1,ESR1,FAT3,FCRL2,FCRL3,GPHN,GPR56,HTR1E,KCNJ1,LGR5,LRRFIP1,LYPD6B,MAGI2,MAP7,MCF2L,MDGA2,MGAM,NTRK3,PDGFRA,PLEK2,PLXNA4,PRAME,PRKCA,PRKCE,PTPRK,PTPRM,PTPRN2,RIMBP2,SCN5A,SEMA5A,SEPT7,SLC22A3,SLC7A14,STXBP6,SYNE1,SYNE2,TLR1,USH2A |
| GO:0016020 | Membrane | 76 | 0.00523 | ADCY9,ANK2,APP,ARHGAP15,ASTN2,ATRNL1,BCL2L1,BNC2,BTN3A1,CADM2,CDC42BPB,CDH11,CDH18,CDH26,CLDN11,CNTNAP2,COL13A1,COL23A1,CPE,CSMD1,DEPDC5,DLG2,DPP10,DPYSL2,EIF3H,ELMO1,EPN1,ESR1,FAT3,FCRL3,GPHN,GPR56,HTR1E,INPP5D,KCNJ1,KIAA1024,LGR5,LINGO2,LRRFIP1,LYPD6B,MAGI2,MAP7,MCF2L,MDGA2,MGAM,MGAT4C,MTMR7,NTRK3,PDGFRA,PLEK2,PLXNA4,PRAME,PRKCA,PRKCE,PTPRK,PTPRM,PTPRN2,RAB11FIP4,RIMBP2,SCN5A,SEMA5A,SEPT7,SLC22A3,SLC7A14,SLC9A9,SORCS2,SPPL2C,STT3A,STXBP6,SYNE1,SYNE2,THSD7B,TLR1,USH2A |
| GO:0044459 | Plasma membrane part | 31 | 0.00523 | ADCY9,ANK2,APP,CLSTN2,CNTNAP2,COL13A1,DLG2,EPB41L3,EPN1,GPHN,GPR56,HTR1E,KCNJ1,LGR5,MAP7,MGAM,NTRK3,PDGFRA,PLEK2,PTPRK,PTPRM,PTPRN2,SEPT7 |
| GO:0033267 | Axon part | 8 | 0.00523 | APP,CNTNAP2,DLG2,DPYSL2,EPB41L3,EPN1,PTPRN2,SEPT7 |
| GO:0044224 | Juxtaparanode region of axon | 3 | 0.00869 | CNTNAP2,DLG2,EPB41L3 |
| GO:0031252 | Cell leading edge | 10 | 0.0126 | CDC42BPB,CNTNAP2,EPB41L3,MCF2L,PHACTR4,PLEK2,PTPRK,PTPRM,SYNE1,SYNE2 |
| GO:0005911 | Cell-cell junction | 10 | 0.0126 | ANK2,APP,CDC42BPB,CLDN11,COL13A1,EPB41L3,MAGI2,PTPRK,PTPRM,SCN5A |
| GO:0031256 | Leading edge membrane | 6 | 0.0236 | CNTNAP2,EPB41L3,PLEK2,PTPRK,SYNE1,SYNE2 |
| GO:0030054 | Cell junction | 18 | 0.0236 | ANK2,APP,BCL2L1,CADM2,CDC42BPB,CLDN11,COL13A1,DLG2,EPB41L3,GPHN,MAGI2,PTPRK,PTPRM,RIMBP2,SCN5A,SYNE1,SYNE2,TP73 |
| GO:0045202 | Synapse | 13 | 0.0236 | ANK2,BCL2L1,CADM2,CLSTN2,DPYSL2,EPB41L3,EPN1,GPHN,MAGI2,PTPRN2,RIMBP2,SEPT7,SYNE1 |
| GO:0031224 | Intrinsic component of membrane | 53 | 0.0237 | ADCY9,ANK2,APP,ASTN2,ATRNL1,BCL2L1,BTN3A1,CADM2,CDH11,CDH18,CDH26,CLDN11,CLSTN2,CNTNAP2,COL13A1,CSMD1,DPP10,ESR1,FAT3,FCRL2,FCRL3,GPR56,HTR1E,KCNJ1,KIAA1024,LGR5,LINGO2,LYPD6B,MDGA2,MGAM,MGAT4C,NTRK3,PDGFRA,PLXNA4,PTPRK,PTPRM,PTPRN2,SCN5A,SEMA5A,SLC22A3,SLC7A14,SLC9A9,SORCS2,SPPL2C,STT3A,STXBP6,SYNE1,SYNE2,THSD7B,TLR1,USH2A |
| GO:0044425 | Membrane part | 58 | 0.0257 | ADCY9,ANK2,APP,ASTN2,ATRNL1,BCL2L1,BTN3A1,CADM2,CDH11,CDH18,CDH26,CLDN11,CNTNAP2,COL13A1,COL23A1,CSMD1,DLG2,DPP10,EPB41L3,EPN1,ESR1,EXOC4,FAT3,FCRL3,GPHN,GPR56,HTR1E,KCNJ1,KIAA1024,LGR5,LINGO2,LYPD6B,MAP7,MDGA2,MGAM,MGAT4C,NTRK3,PDGFRA,PLEK2,PLXNA4,PTPRK,PTPRM,PTPRN2,SCN5A,SEMA5A,SLC22A3,SLC7A14,SLC9A9,SORCS2,SPPL2C,STT3A,STXBP6,SYNE1,SYNE2,THSD7B,TLR1,USH2A |
| GO:0031258 | Lamellipodium membrane | 3 | 0.0288 | PLEK2,SYNE1,SYNE2 |
| GO:0098805 | Whole membrane | 24 | 0.039 | ANK2,APP,BCL2L1,CLSTN2,CNTNAP2,CPE,DEPDC5,DLG2,EPB41L3,EPN1,GPHN,ITGA9,MAP7,MGAM,PLEK2,PTPRN2,RAB11FIP4,SCN5A,SLC7A14,SLC9A9,SYNE1,SYNE2,TLR1,USH2A |
| GO:0016021 | Integral component of membrane | 51 | 0.039 | ADCY9,ANK2,APP,ASTN2,ATRNL1,BCL2L1,BTN3A1,CADM2,CDH11,CDH18,CDH26,CLDN11,CLSTN2,CNTNAP2,COL13A1,CSMD1,DPP10,ESR1,FAT3,FCRL2,FCRL3,GPR56,HTR1E,KCNJ1,KIAA1024,LGR5,LINGO2,MGAM,MGAT4C,NTRK3,PDGFRA,PLXNA4,PTPRK,PTPRM,PTPRN2,SCN5A,SEMA5A,SLC22A3,SLC7A14,SLC9A9,SORCS2,SPPL2C,STT3A,STXBP6,SYNE1,SYNE2,THSD7B,TLR1,USH2A |
| GO:0042995 | Cell projection | 22 | 0.039 | ANK2,CADM2,CLSTN2,CNTNAP2,DPYSL2,EPB41L3,EPN1,EXOC4,GPR56,MAGI2,MCF2L,PDGFRA,PHACTR4,PLEK2,PRKCA,PTPRK,PTPRM,PTPRN2,SYNE1,SYNE2,TSPEAR,USH2A |
| GO:0030424 | Axon | 9 | 0.0452 | CADM2,CNTNAP2,DLG2,DPYSL2,EPB41L3,EPN1,PTPRK,PTPRN2,SEPT7 |

All GO terms with FDR P<0.05 are listed.

figure aligns well with gnomAD European samples, of whom 1.1% are carriers[46]. Thus although behavioural effects are likely to be subtle, and may exert indirect effects within a more complex genetic model as indicated by our gene-environment analyses, the fact that 1 in 100 worldwide may be at risk calls for universal updates to screening protocols.

Beyond this, our study highlights a directionality of effects in which genetically-mediated differences in hearing (directly or indirectly) affect the neuronal development of central auditory processing systems and consequently influence language acquisition. These observations generate two distinct temporal models; (1), the feedback model, in which altered auditory input directly affects neuronal development leading to perceptual deficits that, in turn, increase the risk of speech and language disorders, or (2), the double-hit model, in which altered hearing thresholds combine with existing genetic factors to moderate the risk of speech and language disorders. Exploratory network analyses implicate synaptic connections and cell growth as important processes in hearing-mediated language pathways perhaps suggesting the importance of feedback mechanisms. Importantly, the mouse strains employed here (I129) have a homogeneous background that lacks overt risk mutations. This combines with low *Ush2a* brain-expression[47] and a lack of reported neuronal anomalies in *Ush2a* knockouts[27] to further substantiate the hypothesis that patterns of emergent cochlear output can be shaped by primary stereocilia activity[48,49]. The feedback model fits with a recent single-cell sequencing study, which showed that auditory input during early life can shape gene expression patterns in spiral ganglion neurons (the primary tract between the cochlea and brainstem)[49]. Under this model, early variations in hearing thresholds can have long-lasting and complex downstream effects, presumably through the modification of central mechanisms. The double-hit model aligns with emerging knowledge from high-throughput genomic studies, which indicate the existence of complex shared mechanisms between disorders and further suggest that a "one-gene, one-disorder" expectation represents a gross simplification of genetic mechanisms both in disease and typical development[50,51]. The exact mechanisms by which low-frequency hearing may influence language development remain unclear but given the findings presented here, we propose that differences in auditory input can alter perception of speech. When these occur at critical time-points or are combined with other (as yet unidentified) risk factors, we hypothesise that this may have repercussions for the development of expressive language. The current study considers only air conductance thresholds but future studies may consider other aspects of hearing, for example through the addition of conductive hearing tests or measures of auditory brain responses and exploration of the effects of otitis media. Future investigations are needed to delineate the temporal effects reported here. Such studies will allow us to distinguish between impaired input at the synaptic interface between hair cells and the brain, versus altered linguistic circuitry or feedback, as well as to investigate genetic modifiers and define critical developmental windows for these interactions. The current study directly advances our understanding of the behavioural effects of changes in the *USH2A* gene and indicates that different levels of disruption can target different sound frequencies. Our mouse models further suggests that *Ush2a*-mediated alterations of sound perception can lead to behavioural deficits that extend to vocalization.

## Methods

**Discovery family**. The discovery pedigree consisted of 12 members (Fig. 1). Eight individuals and all descendants of individual I.1 (Fig. 1) were affected by expressive language disorder characterized by acute auditory processing difficulties and speech dysarthria. All descendants of individual I.1 were affected indicating an autosomal dominant inheritance pattern. The family was ascertained through proband IV.1.

**Language phenotype**. The proband (IV.1) was born at full-term by normal delivery following an uneventful pregnancy. There were no early developmental concerns and all gross motor milestones were achieved. However, early language milestones were delayed. First word was reported at 18 months and she was referred to a speech and language therapist at 2 years of age. A diagnosis of Specific Language Impairment (SLI) was given at age 4 years and 8 months. In this assessment, she showed particular difficulties understanding abstract language and linguistic concepts and often failed to follow conversations when no visual cues were given. She had an extensive vocabulary but her language processing was slow and she often showed difficulties finding the word she needed. She showed grammatical difficulties such as sequencing errors, simplification of sentence structure and errors with word structure. On the Children's Communication Checklist (CCC-2), she scored below the 15th percentile on all four language scales (speech, syntax, semantics and coherence) but above this range in scales of inappropriate initiation (60th percentile), use of context (34th percentile), nonverbal communication (42nd percentile) and interests (36th percentile). At a clinical assessment at 58 months, she showed typical hand-eye coordination and performance. She did not present with dysmorphic features and hearing assessments were normal.

There were also concerns regarding the proband's younger Sister's (IV.2) language development. Her first words appeared around the age of 2 years. Motor development was normal. She had mild to moderate bilateral conductive hearing impairment due to recurrent ear infection and grommets were inserted at the age of 3. Following this, hearing assessments were normal but her speech and language difficulties continued and she was diagnosed as having a severe speech disorder in particular with expressive language and dysfluency with very good receptive language skills. The proband (IV.1) and her sister (IV.2) both attend special language units.

The proband's Mother (III.3), maternal Great-Uncle (II.2) and Grandfather (II.1) indicate that the deficits observed in the proband are typical across all family members. The Mother and maternal Great-Uncle have not had formal assessments but both struggled at school requiring speech and language therapy and have difficulties with expressive speech and processing. The similarities between their early difficulties and that of the proband and her sister are striking. Assessment of the Grandfather (II.1) at 62 years of age indicated poor performance across cognitive tasks (verbal and nonverbal) with particular difficulties in tests of recall memory, visual recognition, literacy, executive function and information processing (all below tenth percentile). In contrast, verbal recognition, object naming and auditory attention skills were within the expected range. Pure-tone audiometry showed normal hearing thresholds but deficits were noted across all three tests of central auditory processing (dichotic digits, frequency pattern and duration pattern).

**SNP genotyping**. Seven members of the discovery family (five affected individuals, II.1, II.2, III.3, IV.1 and IV.2, and two unaffected individuals, II.4, III.4, Fig. 1) were genotyped on Illumina HumanOmniExpress-12v1 Beadchips (San Diego, CA, USA; ~750,000 SNPs). SNPs were excluded if the gentrain (genotype clustering quality) score was <0.5 or genotyping success rate was <95%. All individuals had a genotype rate>95%. Genotype data were used to construct haplotype sharing patterns across the pedigree and to call copy number variants (CNVs) as described below.

**Haplotype reconstruction**. SNP genotype data from seven family members were used to construct haplotype sharing patterns within the Merlin package[52]. These data were employed to filter candidate variants from the whole-genome sequence data as described below.

**Copy number calling**. CNVs were called by two separate algorithms; PennCNV[53] and QuantiSNP[54]. All samples had a log R ratio (LRR) SD < 0.35, a B-allele frequency (BAF) drift value <0.002 and a waviness factor between −0.04 and 0.04 in PennCNV and an average LRR SD < 0.3 and BAF SD < 0.15 in QuantiSNP. Any CNV that contained at least three consecutive SNPs, had a confidence value (PennCNV) or log Bayes Factor (QuantiSNP) of >10 and was predicted by both PennCNV and QuantiSNP, with a minimum intersection of 50% each way, was considered to be of 'high confidence'. The innermost boundaries of the two algorithm calls were used. CNVs were excluded if they spanned the centromere or telomeres.

**Whole-genome sequencing**. DNA from two members of the discovery family (II.2 and IV.1, Fig. 1) were subject to whole-genome sequencing enabling the identification of possibly pathogenic variants within shared chromosome regions across the wider pedigree. Sequencing was performed as part of the Oxford University-Illumina WGS500 collaboration (http://www.well.ox.ac.uk/wgs500)[55]. This project includes whole-genome sequences for 156 samples from clinical cases in whom standard genetic tests were negative or where no standard tests were available[55].

Sequencing was completed on the Illumina HiSeq platform (Illumina Inc, San Diego, CA, USA) with 100nt, paired-end runs. Alignment was performed against the Human Reference genome (build 37d5, hg19) in Stampy[56] and duplicate reads removed using Picard (http://broadinstitute.github.io/picard/). Variant sites (Single nucleotide variants and indels less than 50 bp) were called using Platypus (v0.1.8)[57]. The mean depth across all mapped sites was 28.03 and the transtition-transversion ratio across the two samples was 1.99.

Variants that were shared by the two family members and passed quality filters with PHRED quality scores≥20 were identified within vcftools[58] ($N = 17,767$, Supplementary Dataset 1). These were subsequently filtered through a step-wise procedure to include variants which fell within chromosome regions shared only between affected family members (using haplotype reconstruction data from the wider pedigree as described above) (remaining $N = 3743$, Supplementary Dataset 1). Potential functional relevance of shared variants were annotated using SnpEff (v3.2)[59]. Variants that conferred a coding change (frameshift, non-synonymous, canonical splice-variant or stop/start-gain/loss) (remaining $N = 1223$, Supplementary Dataset 1) and were not described (or had a minor allele frequency of 0) in the 1000 Genomes Phase I (v2) data (Apr 2012)[60] (remaining $N = 36$, Supplementary Dataset 1) and dbSNP (build 147)[61] (remaining $N = 6$, Supplementary Dataset 1) were prioritized for follow-up. Variants and filter data are shown in Supplementary Dataset 1. Candidate variants were validated by Sanger sequencing using BigDye (v3.1) on a 3730XL DNA analyzer (Applied Biosystems, California) using standard protocols. Chromatograms were visualized within FinchTV (www.geospiza.com/finchtv).

**Replication cohorts.** Targeted analyses of the identified candidate gene (*USH2A* ± 10Kb - chr1:215786236-216606738, hg19) were performed in two large population cohorts; the Avon Longitudinal Study of Parents and Children (ALSPAC, 7,141 children, 3,615M:3,526F)[41,62] and the UK10K dataset (1646 individuals, 785M:861F)[39]. The ALSPAC population cohort offers a wide range of neurodevelopmental phenotypes (including language, memory, hearing and neuropsychiatric measures) from children born to 14541 mothers from Avon in 1991[41]. In addition to phenotype data, ALSPAC also provides genotype data (Illumina Human Hap 550-quad array) for 8365 children[41] allowing SNP-based association analyses. A subset of ALSPAC children (1867 individuals) had whole-genome sequence data available as part of the UK10K project[39] allowing gene-based association analyses of rare and common variants across the candidate gene.

Both replication cohorts were filtered to include only individuals with available phenotype data, of British ethnicity, born at more than 32 weeks gestation and a birth weight >1500 g. Additional filters were applied for the analysis of common variation in the ALSPAC cohort. These aimed to exclude children with overt pathology that may confound language development, namely nonverbal IQ < 65 and hearing loss (hearing thresholds above 40dbL). After these filters, the ALPSAC replication set included 7141 children (3615M:3526F) and the UK10K replication set included 1681 individuals (806M:875F). The UK10K cohort included fourteen children with heterozygous *USH2A* mutations (Table 2) allowing the consideration of developmental profiles including measures of early language development, later language and cognitive ability, hearing function and neurodevelopmental disorders across carrier children (30 measures in total, Tables 2 and 3).

Analyses targeted three measures of language (early vocabulary, Nonword repetition and Developmental Language Disorder (DLD)) and two measures of air conductance (mid- and low- frequency hearing thresholds) as directed by observations in the heterozygote knockout mice. Details of these measures are provided below and the ALSPAC website contains details of every available measure through a fully searchable data dictionary and variable search tool (http://www.bristol.ac.uk/alspac/researchers/our-data/).

**Early vocabulary (vocab).** The vocabulary measure represents a sum of items that children could use and/or understand, from a list of 123 words, at age 3 (ALSPAC variable KG865). This measure was derived from a parental questionnaire. Data were available for 6165 genotyped children from the ALSPAC cohort and 1614 children from the UK10K cohort. Scores across both datasets ranged from 0 (child did not understand or use any of the 123 words) to 246 (child could use and understand all of the 123 words) (mean = 229.8, SD = 29.4).

**Nonword repetition (NWR).** An adaptation of the Nonword memory test[63] was used to assess short-term memory (ALSPAC variable F8SL105). This measure has been shown to provide an accurate biomarker of speech and language difficulties[64,65]. The tests were completed in clinic and consisted of 12 nonsense words of between 3 and 5 syllables which the child had to listen to and repeat. This test was completed at 8 years of age and data were available for 5229 genotyped children from the ALSPAC cohort and 1572 children from the UK10K cohort. Scores across both datasets ranged from 0 to 12 (mean = 7.3, SD = 2.5).

**DLD status (DLD).** A binary measure of DLD status was defined in line with our previous publications[65,66]; cases performed at least 1 SD below mean on WOLD comprehension (ALSPAC variable F8SL040) OR had CCC verbal fluency AND syntax (ALSPAC variables KU503b and KU504b respectively) >1 SD below mean with no evidence for Autistic Spectrum Disorder (ASD) or hearing impairment.

Typically developing controls were selected to perform above expected levels across all of the three language measures used to define cases (WOLD comprehension, CCC syntax and CCC verbal fluency) and had nonverbal IQ > 80 and presented without neurodevelopmental disorders or special educational needs. The ALSPAC cohort included 731 cases and 2114 controls and the UK10K cohort included 36 cases and 582 controls.

**Mid-frequency hearing (MinMid).** Audiometry was performed as per British Society of Audiologists (BSA) standards—thresholds were taken as 2/3 presentations on the ascending scales. Both air- and bone-conduction were performed using either a GSI 61 clinical audiometer or a Kamplex AD12 audiometer. All hearing tests were carried out by audiologists and trained Staff in a room with minimal external noise (not exceeding 35 dB). Minimum air conduction thresholds were measured for the left and right ears at 0.5, 1, 2 and 4 kHz. Mid-range hearing was defined as the minimum air conduction thresholds across the right and left ears averaged across 1, 2 and 4 KHz (ALSPAC variables F7HS018 and F7HS028, respectively). This measure was available for 4645 genotyped children from the ALSPAC cohort and 1300 children from the UK10K cohort. Thresholds ranged from −8.3 to +40 across these samples.

**Low-frequency hearing (MinLow).** Low-frequency hearing thresholds were defined as the minimum air conduction thresholds across the left and right ears at 0.5KHz. This measure was derived from ALSPAC variables F7HS017, F7HS018, F7HS027 and F7HS028. Data were available for 4563 genotyped children from the ALSPAC cohort and 1277 children from the UK10K cohort. Thresholds ranged from −10 to +40 across these samples.

Separate high-frequency threshold data were not available within the ALSPAC data release for this project.

**Gene-based association analyses.** The UK10K cohort offered genome sequence data, allowing characterization of developmental profiles in identified heterozygous carriers. These sequence data were also employed for gene-based analyses of common and rare variants within RVTESTS[67]. Gene-based testing employed SKAT; a kernel-based method that allows for variants with different directions of effects and can analyze both rare and common variants within a single model[68]. In total, 7691 variants were analyzed. All variants had an allele count of at least 1 in the sample set, affected only single nucleotides (i.e. SNVs), had a minimum mean quality score of 20 and a minimum mean depth of 3 across samples and HWEp > $1 \times 10^{-5}$. The transtition-transversion ratio of the SNVs was 2.2.

**Association analyses of common variants.** SNP data were available for ALSPAC from Illumina 660 and Illumina 550 SNP arrays allowing allelic association analyses of common variants with the PLINK package[42]. Standard quality control procedures[69] were completed on genome-wide SNP data prior to analyses; variants with a minor allele frequency <5%, a call rate of <5%, a Hardy-Weinberg equilibrium $p < 5 \times 10^{-7}$ or a heterozygosity rate more than three standard deviations from the mean were excluded. Per SNP genotype rates were compared between DLD cases and controls and any SNP with a differential missing rate was excluded. Individuals with a genotype rate <95%, discordant sex information or non-Caucasian genetic background were excluded. Following quality control, SNPs across the *USH2A* gene ±10Kb (chr1:215786236-216606738) were pruned using the Tagger algorithm within Haploview[70,71] to derive a pairwise tagging SNP set with R2 < 0.8 consisting of 127 SNPs across 820Kb. Tagging SNPs were analyzed for allelic association within PLINK[42] using a linear model of regression for quantitative traits and a logistic model for discrete traits.

**Genetic interaction analyses.** Gene-environment interaction effects were further modeled within ALSPAC at the gene and genome-wide level using PLINK in which the–interaction command can be used to model SNPxcovariate interactions within a linear regression model ($Y = b0 + b1.ADD + b2.COV1 + b3.ADDxCOV1 + e$)[42]. Gene-level analyses were performed for 12 SNPs across the 5' region of the *USH2A* gene and comprised of three language outcome measures (Early vocabulary (vocab), nonword repetition (NWR) and DLD status) and one interaction factor (Low-frequency hearing threshold (MinLow)). At the genome level, a single outcome measure (Early vocabulary (vocab)) was modelled with a single interaction factor (Low-frequency hearing threshold (MinLow)) for 488205 autosomal SNPs. Manhattan plots were generated using the qqman package[72] within R (v3.4.4) (https://www.r-project.org/).

**Pathway analyses.** SNPs that had P values ≤ $10^{-5}$ in the genome-wide interaction analyses ($N = 450$, Supplementary Dataset 2) were positioned within UCSC (https://genome-euro.ucsc.edu/index.html, hg19) and those which mapped onto known HGNC transcripts ($N = 139$, Supplementary Dataset 2) were entered into pathway analyses to identify over-represented gene classes. Pathway analyses were performed within STRING (https://string-db.org)[73]. Gene ontology classes were analyzed for over-representations using a Fisher exact test with FDR multiple test correction (Table 5). Identified genes were further compared to a list of 37 candidate genes for speech and language-related phenotypes (Taken from ref. [23] and

supplemented with a list from refs. [23,74,75]) and a list of 197 candidate genes for hearing-related phenotypes (taken from http://hereditaryhearingloss.org/, supplemented with a list from the IMPC[75]) (Supplementary Table 2).

**Ush2a mice—subjects**. Six *Ush2a* knockout (KO) male mice[27] were provided by Dr. Jun Yang (University of Utah), and were re-derived on an 129S4/SvJaeJ background strain at the Gene Targeting and Transgenic Facility (GTTF) at UConn Health. All subjects were single housed in standard Plexiglass mouse-tubs (12 h/12 h light-dark cycle), with food and water available ad libitum. F1 subjects were delivered to the University of Connecticut where they were crossed with six wild-type (WT) controls (129S4/SvJaeJ; stock number 009104) purchased from The Jackson Laboratory (Bar Harbor, ME). The resulting F2 offspring were heterozygous (HT) for the *Ush2a* gene, which shows 71% identity with its Human orthologue. Breeding pairs (HT × HT) were used to generate the experimental subjects, such that all genotypes (homozygous knockout, heterozygous, and wild-type) were represented within-litter (F3). F3 genotypes were determined via PCR of earpunch DNA using the following DNA primers: Common (5′-GTGAATACA GGCACCTCTGAATGTGAC-3′), WT (5′-GTCACGGCTGAATCCCGAAGC-3′), KO (5′-GAGATCAGCAGCCTCTGTTCCAC-3′). Twelve WT male mice, 12 HT male mice, and 11 *Ush2a* KO male mice from F3 were randomly selected for behavioural testing as outlined below (12 WT, 11 HT and 11 KO mice were used when recording ultrasonic vocalizations).

**Ush2a mice—auditory processing**. Following puberty, subjects were tested on a battery of auditory processing tasks using a modified prepulse inhibition paradigm which allows free movement during the presentation of sounds that include an unpredictable loud noise burst (see Fitch et al., 2008 for review)[37]. PPI provides a superior index of acoustic processing at higher levels of the central auditory system most relevant to receptive communication. In brief, PPI offers an index of stimulus parameters that are behaviourally detectable, and while simple PPI is brainstem and mid-brain mediated, the use of complex acoustic cues clearly engages auditory cortex[76]. The engagement of cortical/behavioural thresholds is crucial to an ethologically-relevant model of receptive communicative processing. The ability to suppress an acoustic startle response (ASR; an involuntary, reflexive response to an unexpected auditory stimulus [startle eliciting stimulus (SES); 105 dB, 50 ms, broadband white noise burst (1–10 kHz)]) was measured. Subjects were placed on cell-loaded platforms (Med Associates, St. Albans, VT), and presented with varying auditory stimuli generated via RPvdsEx software and a RZ6 multifunction processor (Tucker Davis Technologies, Alachua, FL). Subject motor reflex responses were recorded via a Biopac MP150 acquisition system and Acqknowledge 4.1 software (Biopac Systems, Goleta, CA) connected to the load cell platforms. Tasks are detailed below and included detection of simple tones (15 or 40 kHz) in silence; and of deviant tones (variable duration) in a pure-frequency background (Embedded Tone: 10.5 or 40 kHz background tone, 5.6 or 35 kHz cue tone; Pitch Discrimination: 10.5 or 40.5 kHz tone ± 75 Hz or 8 kHz cue tone). Tone frequencies were determined based on low and high-frequency bounds of the mouse audiogram (~2–50 KHz)[77]. Testing began at postnatal (P) day 65 and continued to P114. Normal Single Tone consisted of 104 trials conducted over one day, where Embedded Tone and Pitch Discrimination each consisted of 300 trials and were conducted over 5 consecutive days. During cued trials, subjects were presented with an auditory cue (prepulse) 50 ms before the presentation of the SES (no cue presentation occurred during uncued trials). If the subject was able to detect the auditory cue, an attenuation (or reduction) of their ASR was expected relative to their ASR during an uncued trial. If the auditory cue was not detected, the response was expected to equate to an uncued trial. Quantification of this phenomenon was termed the "attenuation score" (ATT), which compared the mean amplitude of cued ASR to that of the uncued ASR for each subject, for each session condition.

$$\frac{\text{Mean cued ASR}}{\text{Mean uncued ASR}} \times 100$$

**Normal single tone**. Subjects were first tested on Normal Single Tone (NST) to measure baseline prepulse inhibition, general auditory ability, and to rule out any underlying auditory processing impairments that might impede performance on subsequent auditory processing tasks (i.e. impaired reflex mechanics). Subjects were required to detect a simple single tone (50 ms, 75 dB) against a silent background. This cue was presented 50 ms before the SES on half of the trials (104 cued and uncued trials each, pseudorandom and evenly distributed), at inter-trial intervals (ITI) ranging from 16 s–24 s. Two versions of this task were developed – a 15 kHz version (cue; 50 ms, 75 dB, 15,000 Hz tone) and a 40 kHz version (cue; 50 ms, 75 dB, 40,000 Hz tone). All subjects were able to perform both versions of the task (15 kHz – P65; 40 kHz – P104). The frequency-matched NST score for each subject was used as a covariate in the analysis of further tasks, specifically to eliminate individual differences in PPI or hearing from subsequent auditory processing analyses.

**Embedded tone**. The variable duration Embedded Tone Task (EBT) consisted of 300 pseudorandom trials with ITIs ranging from 16–24 s. Subject's ability to detect a change in tone frequency from a constant pure-tone background was measured,

and ATT scored were calculated. During cued trials, a single cue was presented 100 ms before the SES; for uncued trials, the "cue" was presented 0 ms before the SES (i.e. no cue). Three versions of this task were used: (1) a long-duration EBT task, where the cue duration ranged from 0 ms to 100 ms (cue; 75 dB, 5600 Hz tone & pure-tone background; 75 dB, 10,500 Hz tone); (2) a short-duration EBT task, where the cue duration ranged from 0 ms to 10 ms (cue; 75 dB, 5600 Hz tone and pure-tone background; 75 dB, 10,500 Hz tone); (3) an ultrasonic long-duration EBT task where the cue duration ranged from 0 ms to 100 ms (cue; 75 dB, 35,000 Hz tone & pure-tone background; 75 dB, 40,000 Hz tone). This combination of frequencies and temporal durations was designed to capture the range of processing capacities, allowing us to test for genotype-specific differences in that range. Non-ultrasonic and ultrasonic versions of the task were necessary to determine any Genotype effects observed were frequency dependent. Both non-ultrasonic versions of the task were administered for five consecutive days, and the ultrasonic version of the task was administered for four consecutive days (P67–P78; P103–106).

**Pitch discrimination**. Pitch Discrimination (PD) testing assessed the subject's ability to detect subtle changes in pitch within a constant pure-tone background. Each testing session consisted of 300 pseudorandom trials, with an ITI ranging from 16 s to 24 s. During cued trials, the cue was presented for 300 ms, 100 ms before the SES. "Cues" presented during uncued trials were presented at the same frequency as the pure-tone background. Two versions of this task were used for this study: (1) PD task where the cue frequency deviated 5–75 Hz above or below a 10,500 Hz pure-tone background (cue: 300 ms, 75 dB tone and pure-tone background: 10,500 Hz tone); and (2) ultrasonic PD task where the cue frequency deviated 5–75 Hz above or below a 40,500 Hz pure-tone background (cue: 300 ms, 75 dB tone & pure-tone background: 40,500 Hz tone). A non-ultrasonic PD task was administered for five consecutive days, and an ultrasonic PD task was administered for three consecutive days.

**Ultrasonic vocalizations (USVs; P115-P120)**. Following assessment of auditory processing ability, ultrasonic vocalizations (USVs) were recorded and analyzed using methods adapted from Chabout et al.[78]. Using WT female homecage bedding and urine collected 5 days prior to testing, a single experimental male mouse was placed in a standard Plexiglass tub with a single novel WT female mouse and allowed to freely interact for 5 min. In this setting, a male mouse will vocalize while the female does not, such that recoded calls can be attributed to the male. A Brüel & Kjær Type 4954-B microphone (Brüel & Kjær, Nærum, Denmark), connected to a RME Fireface UC audio interface (RME Audio, Haimhausen, Germany), was placed 5 cm above the top of the Plexiglass tub. USVs were recorded at 192,000 Hz using DIGICheck 5.92 (RME Audio, Haimhausen, Germany) to ensure all USVs were captured. Following USV recording, sound files (.wav) were analyzed in MATLAB (MathWorks) using MUPET (Mouse Ultrasonic Profile ExTraction[79]). Syllables in the range of 35,000 Hz to 110,000 Hz, and duration between 8 ms to 200 ms, were analyzed. If syllables occurred less than 5 ms apart, they were excluded from analyses. Following these parameters, a syllable repertoire was generated, illustrating 40 unique syllables (Supplementary Fig. 1). These 40 unique syllables were then assigned to one of ten potential syllable groups, as defined by Heckman et al.[80]. Eight syllable categories were created; Short, Down-FM, Up-FM, Chevron, Flat, 1-Freq Step, Noisy, and Complex[80]. The mean frequency (kHz) of each syllable was exported from MUPET and used for statistical analyses. Since comparable Genotype effects were seen on all call types, only the mean frequency shift is reported in the text (Fig. 3)—these USV frequency means collapse across all eight call types.

**Statistics and reproducibility (genomic analyses)**. The ALPSAC replication set included 7141 children (3615M:3526F), providing 96% power to detect a variant that explains 0.5% of the trait variance at a Bonferroni-corrected alpha level of $7.87 \times 10^{-5}$. The final UK10K replication set included 1681 individuals (806M:875F) providing 81% power to detect a variant that explains 1% of the trait variance at a Bonferroni-corrected alpha level of 0.0033.

Gene-based analyses were performed within the UK10K dataset. These analyses employed the SKAT test in RVTESTS[67] and considered five traits (as detailed above) using three SNP selection thresholds (all variants, rare variants (MAF ≤ 1%) common variants (MAF ≥ 5%)), yielding a Bonferroni significance threshold of $P = 0.0033$ at an alpha level of 0.05.

SNP-based analyses were performed within the ALSPAC dataset. These analyses employed tests of allelic association within PLINK[42] using a linear model of regression for quantitative traits and a logistic model for discrete traits. Five phenotypes were analyzed across 127 SNPs, yielding a Bonferroni significance threshold of $P = 7.87 \times 10^{-5}$ at an alpha level of 0.05.

Gene-environment interaction effects were modeled within the ALSPAC dataset at the gene and genome-wide level. These analyses used PLINK[42], which employs a linear regression model. Gene-level analyses were performed for 12 SNPs across the 5′ region of the USH2A gene and comprised of three language outcome measures (Early vocabulary (vocab), nonword repetition (NWR) and DLD status) and one interaction factor (Low-frequency hearing threshold (MinLow)) yielding a Bonferroni significance threshold of $P = 0.0014$ at an alpha level of 0.05. At the genome level, a single outcome measure (Early vocabulary (vocab)) was modelled

with a single interaction factor (Low-frequency hearing threshold (MinLow)) for 488205 autosomal SNPs yielding a Bonferroni significance threshold of $P = 1.02 \times 10^{-7}$ at an alpha level of 0.05.

Gene ontology classes were analyzed for over-representations using a Fisher exact test with FDR multiple test correction.

**Statistics and reproducibility (mouse analyses).** Normal Single Tone attenuation scores were analyzed using a one-way analysis of variance (ANOVA) comparing WT, HT, and *Ush2a* KO performance. To account for individual variation in baseline prepulse inhibition and hearing, NST was used a covariate for subsequent statistical analyses (NST 15 kHz was used as a covariate for all non-ultrasonic auditory tasks; NST 40 kHz was used as a covariate for all ultrasonic auditory tasks). EBT and PD tasks were analyzed using a mixed factorial design. Differences in ATT scores for non-ultrasonic EBT 100 and EBT 10 were conducted using a $3 \times 5 \times 9$ repeated measures ANCOVA, with Genotype (three levels; WT, HT, *Ush2a* KO) as the between-subjects variable, and Day (five levels) and cue Duration (nine levels) as the within-subjects variables. Ultrasonic EBT 100 data were analyzed using a $3 \times 4 \times 5$ repeated measures ANCOVA with Genotype (three levels) as the between-subjects variable, and Day (four levels) and cue Duration (five levels) as the within-subjects variables. For Pitch Discrimination, a $3 \times 5 \times 9$ and a $3 \times 3 \times 5$ (for non-ultrasonic PD and ultrasonic PD, respectively) repeated measures ANCOVA was used to determine ATT differences, where Genotype (three levels) was the between-subject variables and Day (three levels, three levels) and Frequency (nine levels, five levels) were the within-subject effects. Statistical analyses were completed used SPSS 24 with an alpha criterion of 0.05.

For the analysis of ultrasonic vocalizations, the overall mean frequency (collapsed across syllable category) was analyzed using a one-way analysis of variance (ANOVA) comparing WT, HT and *Ush2a* KO scores (Fig. 3).

All animal behavioural testing were performed blind to genotype.

**Ethics.** Ethical approval for the discovery family was provided by University of London & St George's University Hospitals. All members provided informed consent/assent of investigation. Ethical approval for ALSPAC was obtained from the ALSPAC Ethics and Law Committee and the Local Research Ethics Committees (http://www.bristol.ac.uk/alspac/researchers/research-ethics/). All animal procedures conformed to the Guide for the Care and Use of Laboratory Animals and were approved by the University of Connecticut Institute for Animal Care and Use Committee (IACUC). The current animal study design adheres to the ARRIVE guidelines[81].

**Reporting summary.** Further information on research design is available in the Nature Research Reporting Summary linked to this article.

## Data availability
All shared variants found in the discovery family are provided in Supplementary Dataset 1. ALSPAC and UK10K SNP and sequence data are available upon application as outlined at http://www.bristol.ac.uk/alspac/researchers/access/. The ALSPAC website additionally contains details of all the data that is available through a fully searchable data dictionary and variable search tool (http://www.bristol.ac.uk/alspac/researchers/our-data/). Auditory processing and USV data from murine behavioral testing are provided in Supplementary Datasets 3 and 4, respectively.

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

## Acknowledgements

We thank Maria Bitner-Glindzicz for her suggestions in the early stages of this work. This manuscript is dedicated to her. We also thank Amanda Hall for her advice regarding hearing phenotypes. The work of the Newbury Lab is currently funded by Oxford Brookes University, the Leverhulme Trust and the Economic and Social Research Council. This work was, in part, completed while Dianne Newbury was at the Wellcome Trust Centre for Human Genetics, Oxford as an MRC Career Development Fellow (G1000569/1). We are extremely grateful to all the families who took part in this study, the midwives for their help in recruiting them, and the whole ALSPAC team, which includes interviewers, computer and laboratory technicians, clerical workers, research scientists, volunteers, managers, receptionists and nurses. The UK Medical Research Council and Wellcome (Grant ref: 102215/2/13/2) and the University of Bristol provide core support for ALSPAC. This publication is the work of the authors and Dianne Newbury will serve as a guarantor for the contents of this paper. A comprehensive list of grants funding is available on the ALSPAC website (http://www.bristol.ac.uk/alspac/external/documents/grant-acknowledgements.pdf). Whole-genome sequencing of the ALSPAC samples was performed as part of the UK10K consortium (a full list of investigators who contributed to the generation of the data is available from www.UK10K.org.uk). ALSPAC GWAS data was generated by Sample Logistics and Geno-typing Facilities at Wellcome Sanger Institute and LabCorp (Laboratory Corporation of America) using support from 23andMe. We thank the WGS500 (see the Supplementary Information for a list of consortium members, including co-author Jenny Taylor) through the High-Throughput Genomics Group at the Wellcome Trust Centre for Human for the generation of the sequencing and genotyping data in the discovery family. The WGS500 project was funded by a Wellcome Trust Core Award (090532/Z/09/Z, Peter Donnelly) and a Medical Research Council Hub grant (G0900747 91070, Peter Donnelly), the NIHR Biomedical Research Centre Oxford, the UK Department of Health's NIHR Biomedical Research Centres funding scheme and Illumina. Animal work was supported by funding from the University of Connecticut Murine Behavioral Neurogenetics Facility, the Connecticut Institute for Brain and Cognitive Sciences (IBACS), and Science of Learning & Art of Communication (NSF Grant DGE-1747486).

## Author contributions

N.L. and A.S. ascertained and assessed the discovery family. R.K., J.T. and D.F.N. generated and analyzed whole-genome and SNP data in the discovery family. A.H. was the

ALSPAC data buddy for this project and compiled and verified all ALSPAC and UK10K datasets for analysis. L.T., H.S.M. and D.F.N. analyzed the ALSPAC and UK10K datasets. P.A.P., A.R.R., A.N.B. and R.H.F. were responsible for all mouse rearing, care, phenotyping and data analysis at the University of Connecticut Murine Behavioral Neurogenetics Facility. P.A.P., L.N., H.S.M., R.H.F. and D.F.N. were responsible for drafting this manuscript. All authors reviewed and agreed the manuscript content. R.H.F. and D.F.N. act as corresponding authors for the murine and Human work respectively.

## Competing Interests

The authors declare no competing interests.
