## [Peer Review File · Communications Biology]

Reviewers' comments:

Reviewer #1 (Remarks to the Author):

In this submission, Perrino et al. report a family in which variants in USH2A are associated with expressive language disturbance. Using a murine model, they demonstrate that heterozygous loss of Ush2a function results in low-frequency hearing loss, and that these mice in turn exhibit higher-order auditory processing. Using cohort data, they hypothesize that distinct USH2A variants may produce both low-frequency hearing loss and expressive language deficits, potentially through gene-environment interactions.

The studied family includes eight individuals with expressive language disorder characterized by impaired fluency, processing speed and ability to follow instructions in distracting auditory environments. A heterozygous stop-gain mutation in USH2A is shared by all affected individuals; in compound heterozygous form, this was previously associated with Usher syndrome and retinitis pigmentosa. We are told that this variant was not felt to cause hearing loss in heterozygotes, but it is not clear whether this prior conclusion was based on detailed audiometric studies.

To investigate the hypothesis that heterozygous loss of USH2A function may be responsible for the language phenotype observed in their family, the authors generate Ush2a knockout mice, in both heterozygous and homozygous form. Homozygous mice exhibited the expected high-frequency hearing loss characteristic of Usher syndrome. Heterozygotes, in contrast, appeared to have more prominent low-frequency hearing loss (at 15Hz). The absence of a gene dose effect is somewhat unexpected, and given the relatively large confidence intervals on the heterozygote measurement, I wonder if this could reflect the effects of sampling and a small cohort size. It would be helpful to know how many replicate experiments were performed on each mouse.

Heterozygous mice were also impaired in pre-pulse embedded tone detection, and pre-pulse pitch discrimination. The authors argue these support an effect of Ush2a loss of function on higher-order auditory processing. It is interesting to note that heterozygotes appear to be most impaired on low-frequency tasks, and that this parallels the effect seen in the Normal Single Tone tests. I wonder if this may reflect an artifact of the covariate adjustment. Additional details on how this was performed would be helpful. In parallel with the expressive language deficits in the index family, Ush2a heterozygous mice vocalized at higher frequencies than wild type mice, and vocalizations were of shorter duration and higher volume. It is unclear from these experiments whether these changes reflect pleiotropic effects of Ush2a dysfunction, or a downstream developmental effect of low-frequency hearing loss. The mechanisms responsible for these changes should be examined in future studies.

To further substantiate a potential effect of USH2A function on language development, the authors identify 14 individuals in the UK10K dataset who possessed 5 separate "pathogenic" (per ClinVar) variants in USH2A. It is not clear whether any of these correspond to the variant found in the index family. Impairments in expressive language and dyslexia were reported for these subjects, and carriers also exhibited impaired low-frequency hearing thresholds. None of these associations appear to be statistically significant based on the reported confidence intervals, though the sample size is small. The presence of dyslexia in these subjects is intriguing, as this would seem to implicate neural pathways beyond auditory processing.

To further support their emerging model, in which USH2A dysfunction leads to low-frequency hearing loss and impaired language development, USH2A SNPs were assessed for association with hearing and early language development in the ALSPAC cohort. There appeared to be modest evidence for

association with low-frequency hearing thresholds. When low-frequency hearing thresholds were included as an interaction factor, a single SNP exhibited moderate association with early vocabulary. While the authors appear to claim support for a gene-environment interaction, a pleiotropic influence of USH2A variants on hearing and language development has not been ruled out.

While we are told that analysis was performed with PLINK, it is unclear how SNPs were QC'd, how covariates were chosen, and whether statistics were adequately corrected for multiple testing. Likewise, additional information on rare variant association testing would be helpful to assess their finding that common variants in USH2A appeared to influence hearing thresholds, whereas rare variants influenced language development.

The association of USH2A variants and heterozygous loss of function with low-frequency hearing loss is interesting, as this expands the clinical spectrum of Usherin variants beyond the high-frequency hearing loss seen in Usher syndrome. Though mechanistically plausible, I find the association with language development to be somewhat less convincing. Reported associations in the ALSPAC cohort are suggestive but not statistically significant. Similarly, the reported evidence for a gene-environment interaction is insufficient to support a strong claim.

Suggestions:

1. It would be helpful to include a few additional clinical details in the body text, namely that some of the affected individuals had normal pure tone audiometry, but evidence of impaired central auditory processing.
2. Were there any unaffected siblings of affected individuals in this family? Were there any individuals with Usher syndrome? If so, it would be helpful to assess these individuals (e.g. through a formal linkage analysis, LOD score) to provide additional evidence for pathogenicity.
3. Please provide additional details on how SNP association and gene-environment analyses were performed. What QC was performed on genotype data? Were variants assessed for missingness, violation of Hardy-Weinberg equilibrium, differential missingness and other technical factors? How were "environmental" covariates chosen? Were reported P values corrected for multiple testing?

Reviewer #2 (Remarks to the Author):

Summary

The major claims of the paper, specifically that an inherited USH2A mutation has an allelic hierarchy affecting low-frequency hearing, are novel and of interest to geneticists, developmental scientists, neuroscientists, speech language pathologists and audiologists. I have raised specific issues that I hope the authors will address below.

Introduction

The introduction does a good job of providing previous context for the Usher syndrome describing the effect of USHA ("gene encodes the Usherin protein which acts as a lateral link between stereocilium, providing structural organization for hair cell bundle development²⁴. Homozygous pathogenic changes in USH2A, and therefore complete absence of the usherin protein, result in disorganization or loss of cochlear outer hair cells²⁵, leading to congenital hearing loss clinically described as Usher Syndrome..." It also clearly mentions the challenges of overcoming behavioral methodologies between humans and mice to validate a behavioral effect of "heterozygote knockout". The following are my concerns and issues for revision.

1. P1. Cite authors who have originally posited the bottom-up models that are described. For

references look for Paula Tallal review articles.

2. P1. Language development occurs well beyond the first few months of life. Either a) be specific about what aspect you are describing as developing in the first few months of life or b) rewrite the sentence to more accurately describe the developmental timeline of language. See April Benasich and Siliva Ortiz-Mantilla for some examples and references.
3. P2. Define what is meant by "robust construct": robust to what?
4. P2. Please specify whether you are describing spoken language or sign etc.
5. P3. Has several distinct concepts. May benefit from separating into two/three paragraphs.
6. Consider specifying the experimental design of the human and mouse experiments.

Methods

1. Overall: Describe and support the statistics used in each section. For example, it is unclear why 2-way ANOVAs were used (HT-WT and WT-HT) instead of 3-way ANOVAs and the use of Cohen's d as a standardized difference without accompanying statistics is not supported.

Results

1. Overall, consider sectioning the results into human and animal studies. As is, the readers start at the family, go to the mouse and then back to children. Although this may be the way that the study logically played out, it is clearer to group by animal.
2. P1. Define stop-gain mutation.
3. P2 and overall. Describe which variables are within and between factors, as well as the effect size or power of each
4. P2 and overall. Consider the use of t-tests to identify group differences more clearly, when appropriate.

Conclusion

The article is consistent with other evidence regarding allelic hierarchy, but also offers a molecular rationalization for why heterozygotes are aphenotypic, opposing indications from different study on hearing disruptions. Additional research is necessary to investigate further into the complexities that the study explored. Despite giving context for the genetic results, the findings are not described in terms of child development. My suggestions are below.

1. Describe how putative low-frequency loss may contribute to language and speech delay.
2. Describe directly how this research has moved the body of scientific knowledge forward

Tables/Figs

Figure 1. Needs labeling of chromatograms and normal abnormal within the figure body. Edit so that someone might understand the figure without looking specifically at the caption.

In general the images and tables were lacking with information in reference to the results and would benefit from more labeled information. For example, it is unclear which groups are significantly different in Fig 2A, should more clearly show that it is an ANOVA.

Reviewer #3 (Remarks to the Author):

Manuscript number: COMMSBIO-19-1554-T

Title: Multi-level evidence of an allelic hierarchy of USH2A variants; hearing, auditory processing and speech/language outcomes.

Summary:

The authors describe a novel genotype-phenotype correlation in a family with a heterozygous mutation in the USH2A gene, causing autosomal dominant severe expressive language disorder. The authors expand this finding into human and mouse model studies to characterize the mechanism. Auditory studies were carried out in heterozygous Ush2a deficient mouse, which showed altered perception of low-frequency stimuli and altered ultrasonic vocalizations. For the human study, two large population cohorts, the Avon Longitudinal Study of Parents and Children (ALSPAC) and UK10K, were investigated and phenotype was compared to the results from the mice studies. The fourteen individuals that were carriers of USH2A mutations in the UK10K cohort showed slightly lower performance on vocabulary and word combination tests, had a higher incidence of speech disturbance, and slightly elevated low-frequency hearing threshold. In the ALSPAC cohort, USH2A polymorphism was associated with language development, when low-frequency hearing threshold was used as an interaction factor.

The authors state that the findings suggest a shared genetic etiology between hearing mechanisms, central auditory processing and language development.

Comments:

- The manuscript was written clear and concise, and was easy to follow.
- The study introduces new insights on the interaction of the auditory function and language development, through genotype-phenotype correlation seen in USH2A mutations in humans and mice. Starting from a single family with USH2A mutation with a unique language phenotype, the authors thoroughly investigated the hypothesis by expanding the mechanistic studies into transgenic mice studies and confirming with large human cohort datasets. The time and effort spent on this project is apparent. The manuscript can be improved by clarification on several points.
- Results. Some abbreviations were not described. What is: CADD, ACMG, PVS1, PS1, PM2, PP1, etc?
- Mice study using the startle reflex. There is no discussion on potential influence derived from vestibular dysfunction or other motor neuron dysfunction in the mutant mice. Also, other conditions that can influence the outcome such as middle ear pathology was not discussed.
- Auditory function analysis in mice. Why were other mainstream tests such as ABR or DPOAE not used in this study? Instinctively, startle reflex test seems to be influenced by many systems other than the auditory system. Is the startle reflex test sensitive enough to tease out such small differences seen in the results?
- How does 15kHz and 40kHz hearing in mice compare with low and high frequency hearing in humans?
- In the pre-pulse detection tests, statistically the attenuation difference seems significant. However, when considering auditory function, is a 5% difference functionally significant? Any discussions?
- Vocalization in mice. The authors suggest that heterozygotes have a higher pitch vocalization because there is low-frequency perceptual deficit. However, based on Figure 3, although the KO mice also have higher pitch vocalization, they have high-frequency hearing loss. Although statistically significant, are these differences really functionally different? It is somewhat difficult to comprehend. More discussions?
- Methods. Mice. What is "I129"? What age mice were used in the study? Why were only male mice used in the study? There is no description on vestibular phenotypes or other motor neuron phenotypes or middle ear pathology regarding the mice.
- Human cohort studies. Hearing threshold was assessed only by air-conduction threshold. Discussion on potential middle ear disease or conductive hearing loss should also be included.
- Although statistically different, is a 1-2dB difference in the low frequency enough to alter language development? Perhaps the findings can be compared to studies done in subjects with otitis media, on language performance?
- Figure 1. Subject III.2. Was this subject deceased? The symbols were not described.
- Table 1. Why was high frequency hearing threshold not considered in the study?

- Results. Cell migration and connection as hearing-modulated pathways, and Table 2. The last paragraph of the manuscript and Table 2 do not fit very well with the rest of the manuscript. This section seems somewhat an over-statement, and is distracting. The information does not strengthen the conclusions from the other sections. Perhaps remove this section altogether?

Response to Reviewer Comments for *Multi-level evidence of an allelic hierarchy of USH2A variants; hearing, auditory processing and speech/language outcomes.*

Peter A. Perrino, MSc, Lidiya Nedevska, MSc, Rose Reader, MSc, Amanda Hill, BSc, Amanda R. Rendall, PhD, Hayley S. Mountford, PhD, Jenny Taylor, Alexzandrea N. Buscarello, BSc, Nayana Lahiri, MD(Res), Anand Sagar, MD(Res), R. Holly Fitch, PhD, & Dianne F. Newbury, DPhil

We would like to thank all the reviewers for their thoughtful and insightful comments. We have detailed responses to these comments below and have revised the manuscript accordingly.

Reviewer #1 (Remarks to the Author):

In this submission, Perrino et al. report a family in which variants in USH2A are associated with expressive language disturbance. Using a murine model, they demonstrate that heterozygous loss of Ush2a function results in low-frequency hearing loss, and that these mice in turn exhibit higher-order auditory processing. Using cohort data, they hypothesize that distinct USH2A variants may produce both low-frequency hearing loss and expressive language deficits, potentially through gene-environment interactions.

The studied family includes eight individuals with expressive language disorder characterized by impaired fluency, processing speed and ability to follow instructions in distracting auditory environments. A heterozygous stop-gain mutation in USH2A is shared by all affected individuals; in compound heterozygous form, this was previously associated with Usher syndrome and retinitis pigmentosa. We are told that this variant was not felt to cause hearing loss in heterozygotes, but it is not clear whether this prior conclusion was based on detailed audiometric studies.

Each of the three previous studies who have identified this variant have involved genetic screens of patients with a clinical diagnosis of Usher syndrome or retinitis pigmentosa. As such, they assumed a recessive model of inheritance which considers heterozygotes as unaffected. They did not directly test hearing in heterozygotes. We have now amended this line (Page 6) to make this point clearer.

To investigate the hypothesis that heterozygous loss of USH2A function may be responsible for the language phenotype observed in their family, the authors generate Ush2a knockout mice, in both heterozygous and homozygous form. Homozygous mice exhibited the expected high-frequency hearing loss characteristic of Usher syndrome. Heterozygotes, in contrast, appeared to have more prominent low-frequency hearing loss (at 15Hz). The absence of a gene dose effect is somewhat unexpected, and given the relatively large confidence intervals on the heterozygote measurement, I wonder if this could reflect the effects of sampling and a small cohort size. It would be helpful to know how many replicate experiments were performed on each mouse.

Text has been added to Pages 16-17 to provide additional information on the auditory processing methodology. Notably, each behavioural mouse task includes multiple trials per day (on some tasks, 100+ trials), and multiple days of testing. Although results are reported as Main effects (across trials/days), the consistent replicability of findings within these repeat samples counters concerns of spurious findings. While it may be unexpected that heterozygous mutations do not result in a gene dose effect, the heterozygous USH2A mutation has been particularly understudied in human samples due to assumptions that the genotype presents as phenotype-free. As such,

further studies exploring how heterozygous USH2A mutations effect Usherin functionality and localization are greatly needed.

Heterozygous mice were also impaired in pre-pulse embedded tone detection, and pre-pulse pitch discrimination. The authors argue these support an effect of Ush2a loss of function on higher-order auditory processing. It is interesting to note that heterozygotes appear to be most impaired on low-frequency tasks, and that this parallels the effect seen in the Normal Single Tone tests. I wonder if this may reflect an artifact of the covariate adjustment. Additional details on how this was performed would be helpful. In parallel with the expressive language deficits in the index family, Ush2a heterozygous mice vocalized at higher frequencies than wild type mice, and vocalizations were of shorter duration and higher volume. It is unclear from these experiments whether these changes reflect pleiotropic effects of Ush2a dysfunction, or a downstream developmental effect of low-frequency hearing loss. The mechanisms responsible for these changes should be examined in future studies.

We agree with the Reviewer that further studies on downstream protein changes resulting from usherin dysfunction are needed, and plan to address these issues in future work. With regards to use of a covariate to assess higher-order behavioural findings in mice, analysis of processing scores on complex acoustic tasks (silent gap in noise (SG), embedded oddball frequency in background frequency (EBT), etc.) used a within-subjects baseline PPI score (single tone in the relevant frequency range in a background of silence) to capture individual differences in auditory sensitivity (gross PPI), or altered peripheral hearing, that might impact higher-order processing indices (ANCOVA, SPSS). Similar methodology accounts for hearing impairments when assessing complex acoustic processing or speech perception scores in human samples. Obviously, if subjects cannot hear stimuli, they will perform poorly on speech processing scores, but this does not imply a speech-specific processing deficit. Since ANCOVA correction depends directly on the correlation between the covariate and the dependent variable, statistical outcomes are affected only to the degree that the baseline PPI scores actually contribute to the variance. Use of this low-level covariate allows us to infer that higher-order processing deficits remain even after peripheral hearing or brainstem acoustic sensitivity differences are accounted for. This is quite important given the absence usherin expression in the brain, and suggests that central acoustic processing changes are an indirect and permanent developmental consequence of peripheral acoustic anomalies in heterozygotes (now clarified on Page 7).

To further substantiate a potential effect of USH2A function on language development, the authors identify 14 individuals in the UK10K dataset who possessed 5 separate "pathogenic" (per ClinVar) variants in USH2A. It is not clear whether any of these correspond to the variant found in the index family.

rs765476745 was not present in the UK10K samples. This is now clarified in the text (Page 8).

Impairments in expressive language and dyslexia were reported for these subjects, and carriers also exhibited impaired low-frequency hearing thresholds. None of these associations appear to be statistically significant based on the reported confidence intervals, though the sample size is small. The presence of dyslexia in these subjects is intriguing, as this would seem to implicate neural pathways beyond auditory processing.

To further support their emerging model, in which USH2A dysfunction leads to low-frequency hearing loss and impaired language development, USH2A SNPs were assessed for association with hearing and early language development in the ALSPAC cohort. There appeared to be modest evidence for association with low-frequency hearing thresholds. When low-frequency hearing thresholds were included as an interaction factor, a single SNP exhibited moderate association with early vocabulary. While the authors appear to claim

support for a gene-environment interaction, a pleiotropic influence of USH2A variants on hearing and language development has not been ruled out.

While we are told that analysis was performed with PLINK, it is unclear how SNPs were QC'd, how covariates were chosen, and whether statistics were adequately corrected for multiple testing. Likewise, additional information on rare variant association testing would be helpful to assess their finding that common variants in USH2A appeared to influence hearing thresholds, whereas rare variants influenced language development.

All SNP data underwent standard quality control procedures (see response to suggestion below). Additional information regarding quality control steps have now been included in the methods section (Pages 15-16). Full details are provided in the Supplementary methods.

The association of USH2A variants and heterozygous loss of function with low-frequency hearing loss is interesting, as this expands the clinical spectrum of Usher variants beyond the high-frequency hearing loss seen in Usher syndrome. Though mechanistically plausible, I find the association with language development to be somewhat less convincing. Reported associations in the ALSPAC cohort are suggestive but not statistically significant. Similarly, the reported evidence for a gene-environment interaction is insufficient to support a strong claim.

Suggestions:

1. It would be helpful to include a few additional clinical details in the body text, namely that some of the affected individuals had normal pure tone audiometry, but evidence of impaired central auditory processing.

Information regarding the hearing assessments have now been added to the main text (Page 6).

2. Were there any unaffected siblings of affected individuals in this family? Were there any individuals with Usher syndrome? If so, it would be helpful to assess these individuals (e.g. through a formal linkage analysis, LOD score) to provide additional evidence for pathogenicity.

The only unaffected individuals in the family were Founder spouses (I.2, II.3, II.4, III.4). This has been clarified in the footnote to Figure 1 (Page 28). A linkage approach was applied to identify shared chromosome regions between seven of the family members. These were used to guide the analysis of the genome sequence data. This has been clarified in methods (Page 15).

3. Please provide additional details on how SNP association and gene-environment analyses were performed. What QC was performed on genotype data? Where variants assessed for missingness, violation of Hardy-Weinberg equilibrium, differential missingness and other technical factors? How were "environmental" covariates chosen? Were reported P values corrected for multiple testing?

Standard quality control procedures (as recommended by Anderson et al 2010, PMID:21085122) were used throughout. For common variants, we excluded SNPs with a minor allele frequency <5%, a call rate of <95%, a heterozygosity rate $\pm 3SD$ from the mean or a Hardy-Weinberg equilibrium $p < 5 \times 10^{-7}$. Individuals with a genotype rate of <95%, discordant sex information or a non-Caucasian genetic background were excluded. SNPs were tested for differential missing rates between DLD cases and controls. Remaining SNPs were pruned for LD. For the gene-based analyses (which included both common and rare variants), all variants had an allele count of at least 1 in the sample set, affected only single nucleotides (i.e. SNVs), had a minimum mean quality score of 20 and a minimum mean depth of 3 across samples and $HWEp > 1 \times 10^{-5}$. Full details of these quality control procedures are in the supplementary methods but an overview has now also been included in the main text (Pages 15-16).

Bonferroni corrections were applied for the number of tests employed through the paper and the significance thresholds adjusted accordingly. Adjusted thresholds are

given in the Table footnotes but have now also been added to the methods (Pages 15-16).

Reviewer #2 (Remarks to the Author):

Summary

The major claims of the paper, specifically that an inherited USH2A mutation has an allelic hierarchy affecting low-frequency hearing, are novel and of interest to geneticists, developmental scientists, neuroscientists, speech language pathologists and audiologists. I have raised specific issues that I hope the authors will address below.

Introduction

The introduction does a good job of providing previous context for the Usher syndrome describing the effect of USHA (“gene encodes the Usherin protein which acts as a lateral link between stereocilium, providing structural organization for hair cell bundle development²⁴. Homozygous pathogenic changes in USH2A, and therefore complete absence of the usherin protein, result in disorganization or loss of cochlear outer hair cells²⁵, leading to congenital hearing loss clinically described as Usher Syndrome...” It also clearly mentions the challenges of overcoming behavioral methodologies between humans and mice to validate a behavioral effect of “heterozygote knockout”. The following are my concerns and issues for revision.

1. P1. Cite authors who have originally posited the bottom-up models that are described. For references look for Paula Tallal review articles.

We have now included reference to Paula Tallal (Page 3)

2. P1. Language development occurs well beyond the first few months of life. Either a) be specific about what aspect you are describing as developing in the first few months of life of b) rewrite the sentence to more accurately describe the developmental timeline of language. See April Benasich and Siliva Ortiz-Mantilla for some examples and references.

We have now rewritten this paragraph to make it more specific (Page 3)

3. P2. Define what is meant by “robust construct”: robust to what?

We refer to the fact that language is robust to individual differences. We have now clarified this in the text (Page 3).

4. P2. Please specify whether you are describing spoken language or sign etc.

We have now clarified this (Page 3)

5. P3. Has several distinct concepts. May benefit from separating into two/three paragraphs.

We have now split this paragraph as suggested (Page 4).

6. Consider specifying the experimental design of the human and mouse experiments.

We have now added this information (Page 5).

Methods

1. Overall: Describe and support the statistics used in each section. For example, it is unclear why 2-way ANOVAs were used (HT-WT and WT-HT) instead of 3-way ANOVAs and the use of Cohen’s d as a standardized difference without accompanying statistics is not supported.

The use of a n ANOVA with a multi-level categorical variable is appropriate when the variable measure is interval or rank order. Examples include drug dosages, age categories (decades), etc. Typically, a single-gene genotype would meet this criteria on the basis of gene dosage (WT, Het, Homozygous). However, in the case of the Ush2a genotype, our data suggest qualitative rather than quantitative differences between the homozygous and heterozygous states. Indeed, on some measures, our results suggest inverse effects at the behavioral level (where our analyses focus). Therefore, a multi-variable ANOVA including all 3 genotype levels would be largely

uninterpretable. On the other hand, individual comparison of each genotype against the WT does meet the requirements for a Between-variable in ANOVA, similar to the use of Sex as a qualitative (rather than quantitative) 2-level Between variable. As far as the use of a Cohen's d, again, this statistic is highly appropriate for the direct planned comparison of 2 (but not multi-level) group means to test specific a priori hypotheses (Encyclopedia of Research Design, Neil Salkind, 2010, Sage Publishing).

Results

1. Overall, consider sectioning the results into human and animal studies. As is, the readers start at the family, go to the mouse and then back to children. Although this may be the way that the study logically played out, it is clearer to group by animal.

The results were written in this order so that the reader could follow the rationale for the targeted investigation of low frequency hearing in as an outcome and interaction factor in the association models. Since the other reviewers stated that the manuscript was easy to follow as written, we have not re-ordered the results section.

2. P1. Define stop-gain mutation.

We have now included this definition (Page 6).

3. P2 and overall. Describe which variables are within and between factors, as well as the effect size or power of each

We have now included information regarding within and between factors (Page 7)

4. P2 and overall. Consider the use of t-tests to identify group differences more clearly, when appropriate.

The Reviewer is correct that Results from between-group comparisons of behavioral data focus on Main effects of Genotype (2 levels). However, we employ repeated measures ANOVA or ANCOVA, with Genotype as a between variable (2-level) rather than using t-tests. This is because all behavioral measures include at least one secondary repeated (within) variable, such as Day, stimulus parameter (duration), syllable type, etc.

Conclusion

The article is consistent with other evidence regarding allelic hierarchy, but also offers a molecular rationalization for why heterozygotes are aphenotypic, opposing indications from different study on hearing disruptions. Additional research is necessary to investigate further into the complexities that the study explored. Despite giving context for the genetic results, the findings are not described in terms of child development. My suggestions are below.

1. Describe how putative low-frequency loss may contribute to language and speech delay.

We propose that differences in auditory input can directly lead to differences in perception which may indirectly influence expressive language development. Further investigations will be required to delineate these relationships and contributory factors. We have now added a paragraph to this effect in the conclusion (Page 13).

2. Describe directly how this research has moved the body of scientific knowledge forward
We have now included a wrap-up sentence to directly describe these advances (Page 14).

Tables/Figs

Figure 1. Needs labeling of chromatograms and normal abnormal within the figure body. Edit so that someone might understand the figure without looking specifically at the caption.

We have now added labelling onto the chromatograms (Page 30).

In general the images and tables were lacking with information in reference to the results and would benefit from more labeled information. For example, it is unclear which groups are significantly different in Fig 2A, should more clearly show that it is an ANOVA.

We have now added additional significance labels to Fig 2A, as well as additional statistics to the Results sections of the text (Page 7)

Reviewer #3 (Remarks to the Author):

Manuscript number: COMMSBIO-19-1554-T

Title: Multi-level evidence of an allelic hierarchy of USH2A variants; hearing, auditory processing and speech/language outcomes.

Summary:

The authors describe a novel genotype-phenotype correlation in a family with a heterozygous mutation in the USH2A gene, causing autosomal dominant severe expressive language disorder. The authors expand this finding into human and mouse model studies to characterize the mechanism. Auditory studies were carried out in heterozygous Ush2a deficient mouse, which showed altered perception of low-frequency stimuli and altered ultrasonic vocalizations. For the human study, two large population cohorts, the Avon Longitudinal Study of Parents and Children (ALSPAC) and UK10K, were investigated and phenotype was compared to the results from the mice studies. The fourteen individuals that were carriers of USH2A mutations in the UK10K cohort showed slightly lower performance on vocabulary and word combination tests, had a higher incidence of speech disturbance, and slightly elevated low-frequency hearing threshold. In the ALSPAC cohort, USH2A polymorphism was associated with language development, when low-frequency hearing threshold was used as an interaction factor. The authors state that the findings suggest a shared genetic etiology between hearing mechanisms, central auditory processing and language development.

Comments:

- The manuscript was written clear and concise, and was easy to follow.
- The study introduces new insights on the interaction of the auditory function and language development, through genotype-phenotype correlation seen in USH2A mutations in humans and mice. Starting from a single family with USH2A mutation with a unique language phenotype, the authors thoroughly investigated the hypothesis by expanding the mechanistic studies into transgenic mice studies and confirming with large human cohort datasets. The time and effort spent on this project is apparent. The manuscript can be improved by clarification on several points.
- Results. Some abbreviations were not described. What is: CADD, ACMG, PVS1, PS1, PM2, PP1, etc?

We have now added definitions for these abbreviations (Page 6)

- Mice study using the startle reflex. There is no discussion on potential influence derived from vestibular dysfunction or other motor neuron dysfunction in the mutant mice. Also, other conditions that can influence the outcome such as middle ear pathology was not discussed.

Homozygous mutations in USH2A cause Usher syndrome type 2 which is characterized by hearing loss at birth and progressive vision loss beginning at puberty. There are no known vestibular abnormalities associated with Usher syndrome type 2 or USH2A mutations (i.e., affected individuals have typical balance and coordination). The current study targeted rapid auditory processing ability and communication ability, however, future studies are necessary to further explore how USH2A mutations impact vestibular and motor systems. Text has been added to Page 5.

- Auditory function analysis in mice. Why were other mainstream tests such as ABR or DPOAE not used in this study? Instinctively, startle reflex test seems to be influenced by

many systems other than the auditory system. Is the startle reflex test sensitive enough to tease out such small differences seen in the results?

Text has been added to Pages 16-17 to clarify the use of PPI instead of more traditional auditory measures, supporting our view that PPI provides a superior index of acoustic processing at higher levels of the central auditory system most relevant to receptive communication. In brief, unlike ABR (which measures electrophysiologic detection of stimuli at the brainstem level), PPI offers an index of stimulus parameters that are behaviorally detectable. And although simple PPI is brainstem and mid-brain mediated, PPI using complex acoustic stimuli clearly engages auditory cortex (Threlkeld, S., Penley, S., Rosen, G.D. & Fitch, R.H. 2008. Auditory gap detection thresholds of intact and microgyric rats following functional deactivation of auditory cortex. NeuroReport, 19, 893 – 898). The engagement of cortical/behavioral thresholds is crucial to an ethologically-relevant model of receptive communicative processing. And while concurrent ABR/cortical AERP would be informative, this would require anesthesia, confounding results. Likewise operant behavioral measures of discrimination (e.g., forced-choice) suffer confounds from training and motivation, potentially biasing discrimination scores (PPI is reflexive and unlearned). For these reasons, PPI offers the ideal measure of concurrent low-level and high-level acoustic processing for the current model.

• How does 15kHz and 40kHz hearing in mice compare with low and high frequency hearing in humans?

Text has been added to Pages 16-17 addressing tone frequencies used. In brief, human hearing ranges from 100 Hz to ~11KHz with optimal hearing in the 2-4KHz range, while the mouse audiogram ranges from 5 to 50Khz with optimal thresholds in the 20 KHz range. As such, 15KHz and 40KHz are suitable markers for low and high frequency hearing in mice, respectively.

• In the pre-pulse detection tests, statistically the attenuation difference seems significant. However, when considering auditory function, is a 5% difference functionally significant? Any discussions?

We argue that mild and low-level hearing losses exert snow-balling effects that can derail higher order communicative processing. These deficits could be further magnified in measures of higher-order language processing in humans — a measure that the mouse model cannot provide. Nonetheless, the implication is that a 5% degradation in complex acoustic processing could, indeed, developmentally derail receptive language processing. This is exactly what is suggested by the complementary human dataset. Text has been added to Page 12 to provide further clarity/discussion.

• Vocalization in mice. The authors suggest that heterozygotes have a higher pitch vocalization because there is low-frequency perceptual deficit. However, based on Figure 3, although the KO mice also have higher pitch vocalization, they have high-frequency hearing loss. Although statistically significant, are these differences really functionally different? It is somewhat difficult to comprehend. More discussions?

Text has been added to Page 8 addressing HT and KO vocalization impairments. We have included a new reference demonstrating that individuals with profound hearing loss (broad-band frequency loss) tend to vocalize at significantly higher vocal frequencies (Mora R, Crippa B, Cervoni E, Santomauro V, Guastini L. 2012. Acoustic features of voice in patients with severe hearing loss. J Otolaryngol Head Neck Surg. 41(1), 8-13). This may reflect a general biologic feature of the mammalian auditory-feedback vocal systems. However, additional research is needed.

• Methods. Mice. What is “I129”? What age mice were used in the study? Why were only

male mice used in the study? There is no description on vestibular phenotypes or other motor neuron phenotypes or middle ear pathology regarding the mice.

Text has been added addressing the background strain on the mice (I129) as well as the age of the subjects (Page 16).

• Human cohort studies. Hearing threshold was assessed only by air-conduction threshold. Discussion on potential middle ear disease or conductive hearing loss should also be included.

The current study took a targeted approach to minimise the number of tests performed but we agree that these would be interesting factors for future studies. We have now added this as an explicit limitation in the discussion (Page 14).

• Although statistically different, is a 1-2dB difference in the low frequency enough to alter language development? Perhaps the findings can be compared to studies done in subjects with otitis media, on language performance?

It is unlikely that these changes will directly lead to language disorder. Instead, we view them as a risk factor whereby mild and low-level hearing loss exert escalating effects that can impede higher-order processing. We have now explicitly stated this in the discussion and drawn a comparison with OME (Page 13).

• Figure 1. Subject III.2. Was this subject deceased? The symbols were not described.

We have now added this to the legend (Page 28).

• Table 1. Why was high frequency hearing threshold not considered in the study?

We specifically targeted low-frequency hearing thresholds in our interaction analyses because of the findings in the mouse heterozygote knockouts and in carriers of pathogenic variants. Direct measures of high-frequency hearing were not available for this cohort. This has now been specified in the methods (Page 15).

• Results. Cell migration and connection as hearing-modulated pathways, and Table 2. The last paragraph of the manuscript and Table 2 do not fit very well with the rest of the manuscript. This section seems somewhat an over-statement, and is distracting. The information does not strengthen the conclusions from the other sections. Perhaps remove this section altogether?

We would argue that this section generalises the results we found for USH2A and allow the exploration of genome-wide effects. Since this point-of-view was not expressed by either of the other reviewers, we would like to leave this section in the manuscript. However, if the editor is also of the opinion that it should be removed, we would be happy to do so.

REVIEWERS' COMMENTS:

Reviewer #1 (Remarks to the Author):

In this resubmission, Perrino et al. present a revision of their investigation into the mechanisms through which USH2A mutation results in expressive language disturbance. Using a murine model, they demonstrate that heterozygous loss of Ush2a function results in low-frequency hearing loss, and that these mice in turn exhibit higher-order auditory processing. Using cohort data, they hypothesize that distinct USH2A variants may produce both low-frequency hearing loss and expressive language deficits, potentially through gene-environment interactions.

The previous review raised identified several points of clarification. These have now been addressed, and the paper now reads more clearly.

The authors have more clearly described their auditory testing methods and subsequent statistical analysis. Genetic quality control and association analysis is also more clearly described. The methods are appropriate, with robust Bonferroni correction being employed throughout.

The paper will be of interest to a wide audience. In its revised form, it should be strongly considered for publication.

Reviewer #2 (Remarks to the Author):

The authors have addressed my concerns. I have no further comments.

Reviewer #3 (Remarks to the Author):

Manuscript number: COMMSBIO-19-1554A

Title: Multi-level evidence of an allelic hierarchy of USH2A variants; hearing, auditory processing and speech/language outcomes.

Comments:

- The authors revised the manuscript according to the reviewers' comments, and is now organized better and details have been clarified. The quality of the manuscript is much improved.
- Page 9. Line 197-201. Mention of heterozygous individuals having low frequency hearing loss, and to see Table S3. I was not able to find this information on TableS3. Perhaps was meant to see a different supplementary table instead?

Reviewer #3 (Remarks to the Author):

Manuscript number: COMMSBIO-19-1554A

Title: Multi-level evidence of an allelic hierarchy of USH2A variants; hearing, auditory processing and speech/language outcomes.

Comments:

- The authors revised the manuscript according to the reviewers' comments, and is now organized better and details have been clarified. The quality of the manuscript is much improved.
- Page 9. Line 197-201. Mention of heterozygous individuals having low frequency hearing loss, and to see Table S3. I was not able to find this information on TableS3. Perhaps was meant to see a different supplementary table instead?

As requested by reviewer 3, supplementary table 2 includes low-frequency hearing data (although see note below regarding the movement of supplementary tables to the main text. Supplementary Table 2 is now Table 2 in the main text).